# A chemically inert bismuth interlayer enhances long-term stability of inverted perovskite solar cells

Shaohang Wu [1], Rui Chen[1], Shasha Zhang[1], B. Hari Babu[2], Youfeng Yue[3], Hongmei Zhu[1], Zhichun Yang[1], Chuanliang Chen[1], Weitao Chen[1], Yuqian Huang[1], Shaoying Fang[1], Tianlun Liu[1], Liyuan Han[4] & Wei Chen[1,2]

Long-term stability remains a key issue impeding the commercialization of halide perovskite solar cells (HPVKSCs). The diffusion of molecules and ions causes irreversible degradation to photovoltaic device performance. Here, we demonstrate a facile strategy for producing highly stable HPVKSCs by using a thin but compact semimetal Bismuth interlayer. The Bismuth film acts as a robust permeation barrier that both insulates the perovskite from intrusion by undesirable external moisture and protects the metal electrode from iodine corrosion. The Bismuth-interlayer-based devices exhibit greatly improved stability when subjected to humidity, thermal and light stresses. The unencapsulated device retains 88% of its initial efficiency in ambient air in the dark for over 6000 h; the devices maintain 95% and 97% of their initial efficiencies after 85 °C thermal aging and light soaking in nitrogen atmosphere for 500 h, respectively. These sound stability parameters are among the best for planar structured HPVKSCs reported to date.

[1] Wuhan National Laboratory for Optoelectronics, Huazhong University of Science and Technology, Luoyu Road 1037, 430074 Wuhan, China. [2] Shenzhen Key Laboratory of Nanobiomechanics, Shenzhen Institutes of Advanced Technology, Chinese Academy of Sciences, Shenzhen, 518055 Guangdong, China. [3] Electronics and Photonics Research Institute, National Institute of Advanced Industrial Science and Technology (AIST), 1-1-1 Higashi, 305-8565 Tsukuba, Japan. [4] State Key Laboratory of Metal Matrix Composites, School of Materials Science and Engineering, Shanghai Jiao Tong University, 200240 Shanghai, China. These authors contributed equally: Shaohang Wu, Rui Chen, Shasha Zhang. Correspondence and requests for materials should be addressed to L.H. (email: han.liyuan@sjtu.edu.cn) or to W.C. (email: wnlochenwei@mail.hust.edu.cn)

Highly efficient and low-cost halide perovskite solar cells (HPVKSCs) are regarded as one of the most promising photovoltaic technologies to realize commercialization in the near future[1–5]. However, the long-term stability of HPVKSCs under real working conditions, involving stresses from moisture, heat, light, and the electric field, is still a challenge that must be addressed[6].

The irreversible degradation of HPVKSCs can be mainly attributed to three phenomena: (1) volatilization of the organic components in organic–inorganic hybrid perovskites[7], especially for the benchmark methylammonium lead triiodide (MAPbI$_3$) under thermal aging conditions above 85 °C[8], (2) permeation of external H$_2$O/O$_2$ and its induced degradation of perovskites, and (3) reactions between the normally used metal electrodes and halogens from the perovskites during prolonged operation[9,10]. For the last two issues, especially the third one, their resolutions are largely related to the essential properties of the interfacial layers (i.e., morphologic pin-hole free, chemical resistivity to H$_2$O/O$_2$ and halides), which could act as robust permeation barriers to block the undesired processes. If the interfacial layers are permeable or have pin-holes, sufficient separation between the perovskites and electrode metals cannot be achieved, and the corrosive reaction could be fast. For example, in regular-structured HPVKSCs, Grätzel et al. observed the inward diffusion of Au into the perovskite layer through Spiro-OMeTAD under thermal aging at 70 °C[10], which may introduce broad tail states in the perovskite layer and thus increase the undesirable non-radiative recombination in the devices[11]. A similar case has been proven to exist in inverted-structured HPVKSCs. Wang et al. observed that I$^-$ ions could diffuse through the [6,6]-phenyl-C61-butyric acid methyl ester (PCBM) layer and react with the Ag electrode under thermal aging at 85 °C in a N$_2$ atmosphere. This reaction could facilitate the escape of I$^-$ ions from the perovskites and thus lead to an accelerated decomposition of the perovskite film, which results in an irreversible degradation of the device performance[12].

Thus, a robust interfacial barrier is essential, and barrier design has been proven to be a feasible solution to the degradation of HPVKSCs that focuses on hindering the adverse diffusion of ions/molecules, such as the outward diffusion of perovskite species and inward diffusion of metal ions from the electrode or H$_2$O/O$_2$ from the atmosphere[13–16]. For example, multilayer interfacial layers made of PDCBT/Ta-WO$_x$[17], Spiro-OMeTAD/MoO$_x$[18], Spiro-OMeTAD/MoO$_x$/CuPc[19], EH44/MoO$_x$[20], and CuSCN/graphene[21] were designed to strengthen the penetration barriers of regular-structured HPVKSCs and were considered essential for the largely enhanced long-term stability. In inverted-structured HPVKSCs, the electron transport layers (ETLs) are normally composed of fullerene derivatives such as PCBM or C$_{60}$ and a buffer layer such as bathocuproine (BCP)[22], LiF[23], nano-carbon materials[13], metal acetylacetonates[24], and amorphous or nanocrystalline metal oxides[23,25–29]. Many bilayer-structured ETLs have also been found to be essential for selective charge extraction and enhancing the device's power conversion efficiency (PCE) and long-term stability. In particular, Riedl et al.[15] reported an impermeable triple-layered ETL composed of PCBM/Al-doped ZnO (AZO)/SnO$_x$, in which the SnO$_x$ layer deposited by atomic layer deposition (ALD) technique possesses an extremely low water vapor transmission rate of approximately $7 \times 10^{-5}$ g m$^{-2}$ day$^{-1}$. Their unencapsulated MAPbI$_3$-based devices (MA-HPVKSCs) exhibited no degradation under thermal aging at 60 °C for 1000 h. Recently, McGehee et al. even found that their MA-HPVKSCs did not decay after 1000 h at 85 °C in the dark when a similar ALD-deposited SnO$_2$ barrier with a chemically inert ITO electrode was implemented[14]. More impressively, Park et al.[16] recently reported that Cs$_{0.05}$MA$_{0.95}$PbI$_3$-based HPVKSCs

with ALD-AZO barrier layers retained 86.7% of their initial efficiency after 500 h of maximum power point (MPP) tracking under continuous 1 Sun illumination at 85 °C in ambient air.

To date, all of the reported barrier layers have been made of non-metals, such as carbon-based materials (such as graphene, nano-carbon materials) or metal oxides. The use of metallic materials in stable perovskite-based devices is challenging because of the "metals–halides perovskites" reaction. McGehee et al. even recommended replacing the metal electrodes with transparent conducting oxides, such as ITO, for stable perovskite-based devices[30]. In view of the above, when examining the periodic table of elements, we still find potential in the region between the metals and non-metals, i.e., bismuth (Bi) and antimony (Sb), which are referred to as semimetals.

Bi was chosen mainly because it prevents corrosion of halide perovskites, which is in contrast to most of the currently used metal electrodes, such as Ag, Au, Al, and Cu[9,10,31–33]. The tendency of the corrosion reaction can be predicted by theory, as discussed later. Previous experimental results indicated that Bi required a temperature of at least 150 °C to trigger a chemical reaction with gaseous iodine to form bismuth (III) iodide[34]. Furthermore, the rhombohedral lattice of Bi has crystal unit parameters of $|\mathbf{a}| = |\mathbf{b}| \approx 4.5$ Å and trigonal axis $|\mathbf{c}| \approx 11.8$ Å[35–38], resulting in anisotropic film growth and leading to a flaky structure with limited grain boundaries, which is essential for the as-formed Bi interlayer to act as a robust penetration barrier. Additionally, Bi can be easily deposited by conventional thermal evaporation, unlike other anti-corrosive metals such as molybdenum (Mo). To achieve a vapor pressure of $10^{-4}$ Torr for thin film deposition, the required temperature for Bi is 517 °C, which is lower than that required for Ag (832 °C) and much lower than that for tungsten (W, 2757 °C) and Mo (2717 °C)[39]. Mo and W films typically must be deposited by magnetron sputtering or e-beam evaporation. These high-energy processes may easily damage the perovskite layers underneath the electrodes. In addition, Bi is cost-effective, whose cost is approximately 1/10 that of Ag[40].

Here, we demonstrate a facile processing strategy for highly stable HPVKSCs with the structure FTO/Li$^+$-doped Ni$_x$Mg$_{1-x}$O (NiMgLiO)/Perovskite (PVK)/PCBM/BCP/Bi/Ag by introducing the compact Bi interlayer. The devices and films were characterized by photoluminescence (PL), scanning electron microscopy (SEM), X-ray photoelectron spectroscopy (XPS), X-ray diffraction (XRD), time of flight secondary ion mass spectrometry (ToF-SIMS), etc., which demonstrate that the Bi interlayer acts as a robust penetration barrier that blocks the unfavorable external molecules from diffusing into the devices and suppresses the diffusion of ions between the metal electrode and the perovskites. As a result, our unencapsulated MA-HPVKSCs retain 88% of their initial efficiency after being maintained in the dark in ambient air without humidity control for 6000 h. During long-term light soaking or thermal aging at 85 °C in the dark, the devices with Bi interlayers are considerably more stable than devices without Bi. This Bi-interlayer strategy is easily scalable and highly repeatable and may pave the way to the future practical application of large-area and highly stable HPVKSCs.

## Results

**Chemically inert characteristic of Bi film**. Here, regarding the chemically inert characteristic of Bi, we primarily focus on its anti-corrosivity to halide perovskites. To demonstrate this key characteristic, MAPbI$_3$/metal samples were prepared by evaporating 5-nm-thick Bi, Cu, Au, or Ag films (which should have island-like morphology) directly onto MAPbI$_3$ films, and XPS characterization was performed on both the fresh and aged

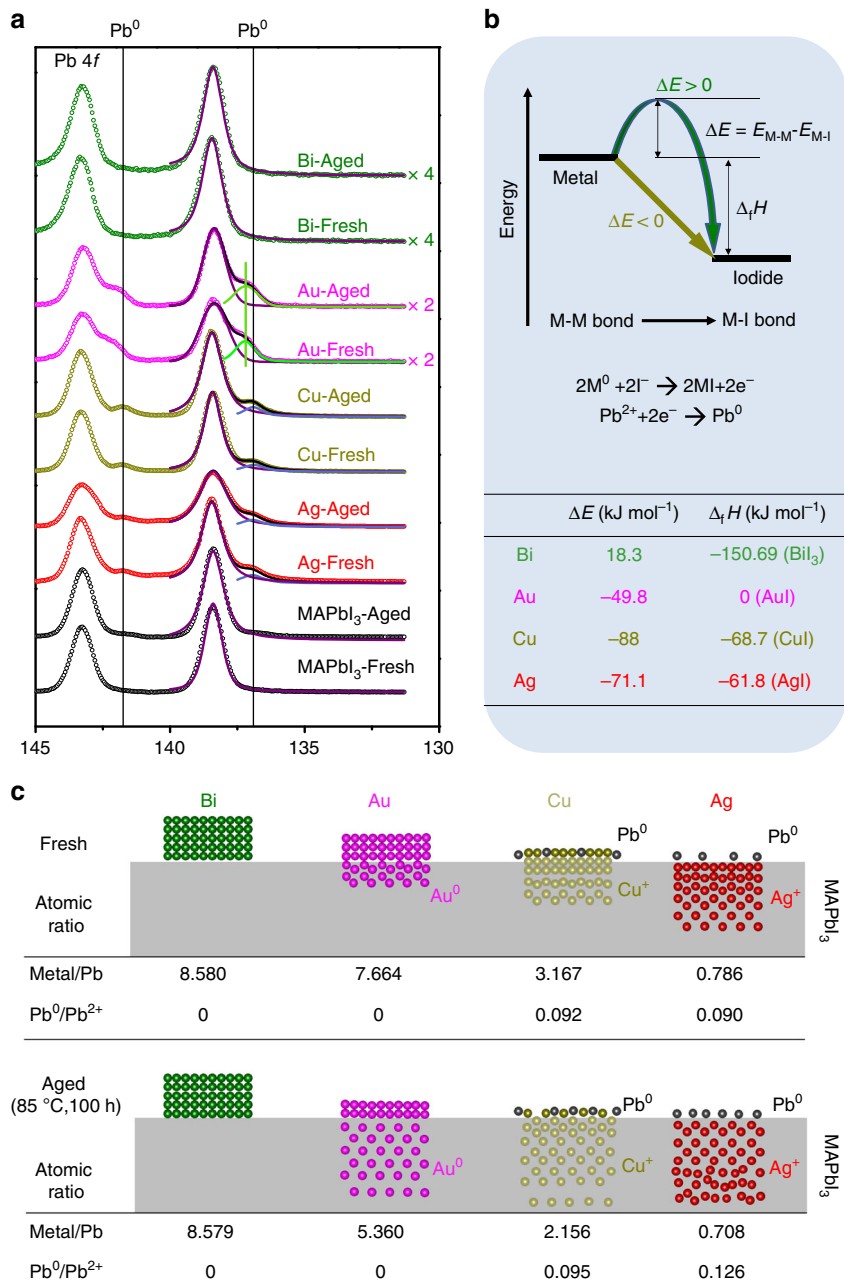

**Fig. 1** Chemical reaction of metal-perovskite. **a** XPS spectra of Pb 4f for fresh and aged samples, including MAPbI$_3$/Bi, MAPbI$_3$/Au, MAPbI$_3$/Cu, MAPbI$_3$/Ag, and only MAPbI$_3$ (aging conditions: in the dark, in N$_2$ atmosphere, 85 °C, 100 h). **b** Schematic diagram and the corrosion reaction equations. The standard enthalpy of formation ($\Delta_f H$, 298.15 K) and the reaction barrier ($\Delta E$) are also listed. **c** Schematic diagram of metal diffusion into the perovskite and the atomic ratios at the upper surface. Source data are provided as a Source Data file

samples (aging conditions: in the dark and N$_2$ atmosphere, at 85 °C for 100 h). As shown in the Pb 4f XPS spectra (Fig. 1a), peaks at 138.40 and 143.25 eV can be attributed to the Pb–I bonds in perovskite, and peaks at 136.90 and 141.75 eV are from metallic Pb$^0$ [32]. The figure shows there is no metallic Pb$^0$ found in the purely MAPbI$_3$ film samples even after thermal aging, indicating that MAPbI$_3$ may remain stable when it is not attacked by the active metal films. However, when Cu or Ag was evaporated onto MAPbI$_3$, the metallic Pb$^0$ appeared, combined with the dominant Ag$^+$ and Cu$^+$ peaks in Ag 3d and Cu 2p spectra (Supplementary Figure 1), which demonstrate the corrosion reaction between Cu/Ag and MAPbI$_3$. For MAPbI$_3$/Au, although there are no peaks corresponding to metallic Pb$^0$ or Au$^+$,

shoulders are observed at 137.2 and 142.2 eV. These were suggested to be attributed to partial charge transfer at the MAPbI$_3$/Au interface in other work, which also gave rise to stability problem for Au electrode-based HPVKSCs[31,32,41]. In contrast, for MAPbI$_3$/Bi, neither additional Pb 4f peaks nor Bi$^{3+}$ peaks were observed, indicating Bi is chemically stable on the MAPbI$_3$. This feature is unique in comparison to most of the metal electrodes reported to date.

A thermodynamic analysis provides insight into the mechanism of the reactions between the perovskite and the metals. It is known that the corrosion reaction between metals and halogens can be considered a process of breaking the elemental metal–metal bonds and forming the corresponding

metal–halogen bonds. When the formation enthalpy of metal iodide is not negative or there is an energy barrier for the change of chemical bond (the barrier can be denoted as $\Delta E$, equal to the energy difference between the metal–metal bond ($E_{M-M}$) and the metal–halogen bond ($E_{M-I}$), $\Delta E = E_{M-M} - E_{M-I}$), the corrosion reaction of metal to metal iodide is difficult or impossible (Fig. 1b, and see the formation enthalpy and energy barrier in Supplementary Table 1). In this work, the formation enthalpy of AuI is not negative, thus there was no metallic $Pb^0$ or $Au^+$ in the $MAPbI_3$/Au samples (more information about Au–I case in Supplementary Note 1). To the contrary, AgI ($\Delta_f H = -61.8$ kJ mole$^{-1}$, $\Delta E = -71.1$ kJ mole$^{-1}$) and CuI ($\Delta_f H = -68.7$ kJ mole$^{-1}$, $\Delta E = -88$ kJ mole$^{-1}$) can be formed easily and are stable, so $Pb^0$ and $Ag^+/Cu^+$ were observed in their corresponding samples. For Ag and Cu samples, we speculate that the following corrosion reactions may occur:

$$2M^0 + 2I^- \rightarrow 2MI + 2e^- \qquad (1)$$

$$Pb^{2+} + 2e^- \rightarrow Pb^0 \qquad (2)$$

Although Bi has a negative $\Delta_f H$, there is an energy barrier of 18.3 kJ mole$^{-1}$ hindering the formation of $BiI_3$. This is similar to the case of C ($\Delta E = 365.2$ kJ mole$^{-1}$, $\Delta_f H(CI_4) = -392.9$ kJ mole$^{-1}$) and Mo ($\Delta E = 168.6$ kJ mole$^{-1}$, $\Delta_f H(MoI_2) = -103.9$ kJ mole$^{-1}$), which have been proven to be stable in previous experiments and theory[31,42,43].

Also, the corrosion reactions could proceed further during thermal aging (Fig. 1c). In the $MAPbI_3$/Ag and $MAPbI_3$/Cu samples, the $Pb^0/Pb^{2+}$ atomic ratios increased after 85 °C thermal aging for 100 h, from 0.090 to 0.126 for $MAPbI_3$/Ag, and from 0.092 to 0.095 for $MAPbI_3$/Cu. Furthermore, the thermal aging appears to have promoted the penetration of Ag, Cu, and Au ions into the $MAPbI_3$ films, resulting in reduced atomic ratios at the sample surface (XPS detection depth of about 10 nm), from 0.786 to 0.708 for Ag/Pb, from 3.167 to 2.156 for Cu/Pb, and from 7.664 to 5.360 for Au/Pb. In fact, a recent density function theory (DFT)-based computation study also shows that Ag, Cu, and Au easily diffuse into perovskites because of their very low diffusion barriers[31]. In contrast to Ag, Cu, and Au, the atomic ratios of Bi/Pb in the fresh and aged $MAPbI_3$/Bi samples almost remained unchanged at 8.580 and 8.579, respectively, indicating that there is very limited penetration of Bi into the $MAPbI_3$ films during thermal aging.

The chemical inertness of Bi is also applicable to other perovskites. The thermal stability of perovskites/Bi (40 nm) was studied by XRD characterizations of different perovskites, including $MAPbI_3$, $(FAMA)Pb(IBr)_3$, $(FACs)Pb(IBr)_3$, and $(FAMACs)Pb(IBr)_3$, and taking Ag and Cu as the references. The resulting optical images (Supplementary Figure 2) and XRD patterns (Supplementary Figure 3) show that all fresh samples show metallic luster and strong perovskite characteristic peaks at approximately 14°, especially for Bi and Cu. After thermal aging (85 °C, ambient air, in the dark) for 48 h, only the Bi samples maintained their metallic luster, while the Ag samples and Cu samples lost their metallic luster, indicating severe degradation of the perovskites and corrosion of the metals. From the XRD results (Supplementary Figure 3), much weaker peaks of the perovskite phases and stronger peaks of $PbI_2$ were observed for the aged Ag and Cu samples, and for some samples, the perovskite peaks disappeared completely. However, the XRD patterns of the Bi sample changed much less with aging. These results consistently indicate that Bi is chemically inert for these kinds of perovskites, even under harsh conditions.

**Morphology and structural impermeability of the Bi film.** Having demonstrated that Bi is chemically inert under these conditions, we prepared Bi films with different thicknesses on the surfaces of Glass/PCBM/BCP substrates by a simple thermal evaporation deposition. Figure 2a shows SEM images of the as-deposited Bi films on the Glass/PCBM/BCP substrates. The image shows that the 5-nm-thick sample is composed of many nanometer-scale islands, indicating poor coverage on the film surface. On the other hand, the 10-nm-thick sample is comprised of some flakes along with nanoparticles filling in the gaps between flakes. When the film thickness increased to 20 nm, a film with completely continuous coverage showing closely-packed flakes (with the average size of approximately 110 nm) was achieved. A further increase in the thickness results in a growth in the size of Bi flakes to approximately 190 nm. Next, we characterized the crystalline structures of the Bi films by XRD. The XRD patterns of Bi films on glass/PCBM/BCP substrates are shown in Fig. 2b. The three main peaks are attributed to (003), (006), and (009) planes of rhombohedral Bi (JCPDS 85-1331)[37–40], and their intensities increase with increasing Bi thickness. This indicates an anisotropic growth of the Bi film with predominant orientation along the c-axis of the rhombohedral crystal unit, as shown in Fig. 2c. The strong and sharp peaks reveal high structural orientation and crystallinity of the Bi film, which is consistent with the film morphology (Fig. 2).

The high structural orientation gives the Bi layer a very compact morphology and perfect impermeability. To examine the impermeability of the Bi interlayer, we designed a qualitative experiment to test its water/oxygen vapor transmission rate, as shown in Supplementary Figure 4. Large-area Ca films (thickness = 60 nm, area = $3 \times 3$ cm$^2$) on glass substrates covered by different interfacial layers including PCBM (60 nm)/BCP (5 nm), PCBM (60 nm)/BCP (5 nm)/Ag (20 nm), and PCBM (60 nm)/BCP (5 nm)/Bi (20 nm) were compared under controlled air conditions (25 °C and 30% relative humidity (%RH)). By monitoring the color change of the deposited Ca films, we could qualitatively judge the impact of different interfacial layers on the water/oxygen vapor transmission rates. From the optical images, the bare Ca film without interfacial layers quickly reacted with $H_2O/O_2$ from the air and turned into colorless CaO or $Ca(OH)_2$, whereas the hydrophobic interfacial layer of PCBM (60 nm)/BCP (5 nm) retarded this process by about 1 h. Unlike the first two samples, an additional Ag electrode layer (20 nm) on top of the PCBM/BCP strengthened the penetration barrier against $H_2O/O_2$, however, the shielding capability was not sufficient. After 3 h, the Ca film was seriously damaged because the as-deposited Ag film was comprised of nanocrystals and rich in grain boundaries (having a disconnected nature) that allowed $H_2O/O_2$ to penetrate (Supplementary Figure 5a). On the contrary, a Bi layer (20 nm) on top of the PCBM/BCP (with a dense film morphology and limited grain boundaries (Supplementary Figure 5b)) presented a much more robust penetration barrier to $H_2O/O_2$. After 3 h, the PCBM (60 nm)/BCP (5 nm)/Bi (20 nm) sample maintained its initial color without any change across the entire film's surface. Thus, as the penetration barrier for $H_2O/O_2$, Bi performs much better than Ag thanks to its highly oriented growth and crystal structure, which differed greatly from Ag (Supplementary Figure 5c-f). Furthermore, it should be noted that such a robust penetration barrier can be easily scaled up to a large area of more than the $3 \times 3$ cm$^2$ used in this experiment.

**Device fabrication and photovoltaic performances.** HPVKSCs were fabricated with Bi interlayers in an inverted structure of FTO/NiMgLiO/PVK/PCBM/BCP/Bi/Ag (Fig. 3a). The studied HPVKSCs had large active areas (typically 1 cm$^2$). The

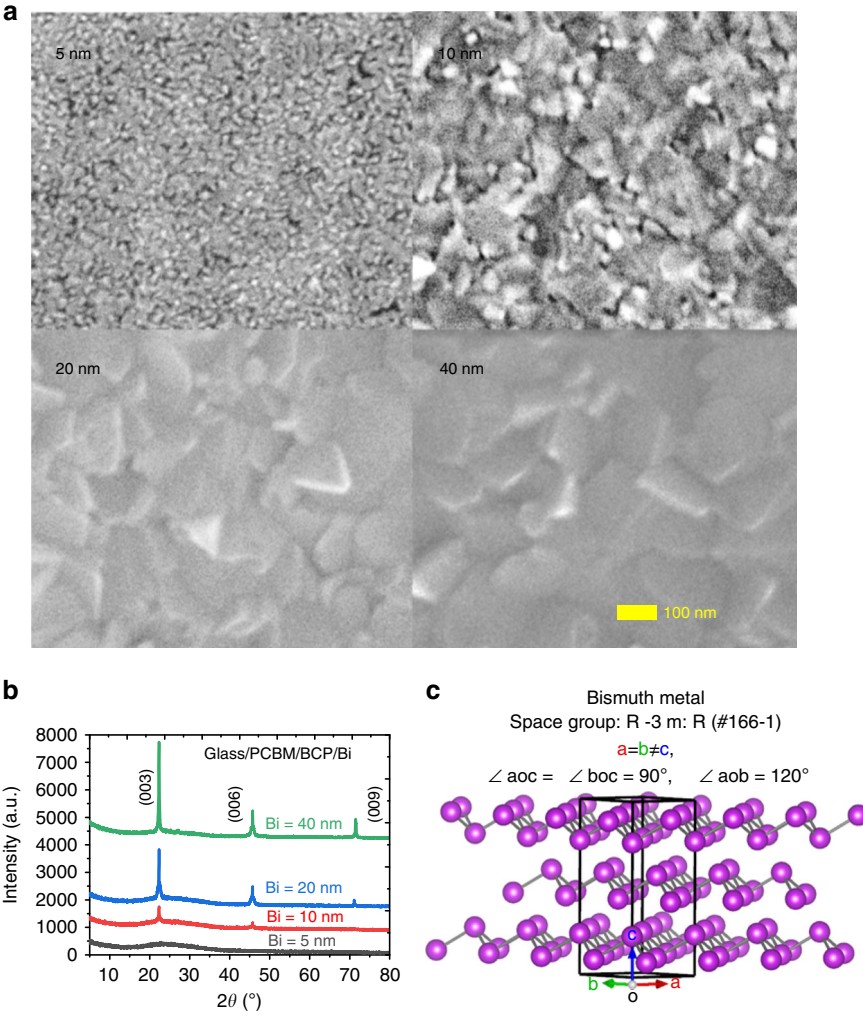

**Fig. 2** Structural properties of the bismuth film. **a** SEM images of the as-deposited Bi films on Glass/PCBM/BCP substrates with different thicknesses (scale bar: 100 nm); **b** XRD patterns of the as-deposited Bi films on Glass/PCBM/BCP substrates with different thicknesses; and **c** schematic diagram of the crystal structure of Bi. Source data are provided as a Source Data file

corresponding band energy diagram and the SEM cross-sectional image of a whole MA-HPVKSCs are portrayed in Fig. 3b, c, respectively. The corresponding band energy levels were obtained from the literature[22,23]. The benchmark MAPbI$_3$ film was prepared with an average thickness of approximately 400 nm by using the classic anti-solvent extraction method, and this layer was sandwiched between 20-nm-thick p-NiMgLiO and 60-nm-thick n-PCBM charge extraction layers prepared according to our previous work[23]. An ultrathin (5 nm) BCP buffer layer was deposited by thermal evaporation and was inserted between the PCBM and the metal electrode to form a remarkable Ohmic contact to suppress the charge accumulation between them[22]. The Bi interlayer was deposited by adopting similar technology and sandwiched between the BCP and Ag (150 nm). Here, the Bi interlayer performs as a continuous blocking layer for the diffusion of unfavorable molecules/Ag ions, but at the same time it should not affect the transfer of photon-generated charges in the devices. The presence of the continuous Bi interlayer could be observed from the SEM cross-sectional image in Fig. 3c. In accordance with previous reports[44], the work function of the Bi layer was estimated to be 4.25 eV, which is close to that of Ag (4.30 eV) (Supplementary Figure 6 and Supplementary Table 2). Therefore, the contact of the PCBM/BCP/Bi/Ag architecture should be as good as that of the PCBM/BCP/Ag architecture.

The conductivity of Bi is roughly 80 times lower than that of Ag; however, it is comparable to those of transparent conductive oxides, such as indium tin oxide[45]. The well-matched energy level along with the good conductivity implies that the Bi interlayer with the optimized thickness will have a negligible negative impact on the charge transfer in HPVKSCs. Photocurrent density–voltage ($J$–$V$) curves were measured for devices with different thicknesses of Bi interlayers (see their $J$–$V$ curves and IPCEs in Supplementary Figure 7, and their photovoltaic parameters in Supplementary Table 3). The devices with a Bi interlayer less than 20 nm show good photovoltaic performance; however, a thicker Bi interlayer (40 nm) is found to be harmful to the device's efficiency. This is due to the increased series resistance ($R_S$). From the AFM and cross-sectional SEM images (Supplementary Figure 8), it was found that the surface morphology of Bi significantly changed as its thickness increased from 10 to 80 nm. Some protuberant columnar crystals with heights greatly exceeding the designed film thickness were clearly observed in the 40-nm-thick and especially in the 80 nm-thick Bi films, corresponding to their increased surface roughness ($R_a$) from 3.652 to 37.672 nm, which may lead to increased contact resistance between the Bi and Ag[46]. The contact resistance may be the original reason for the unexpectedly larger $R_S$ for the HPVKSC with 40-nm-thick Bi. Figure 3d shows typical $J$–$V$

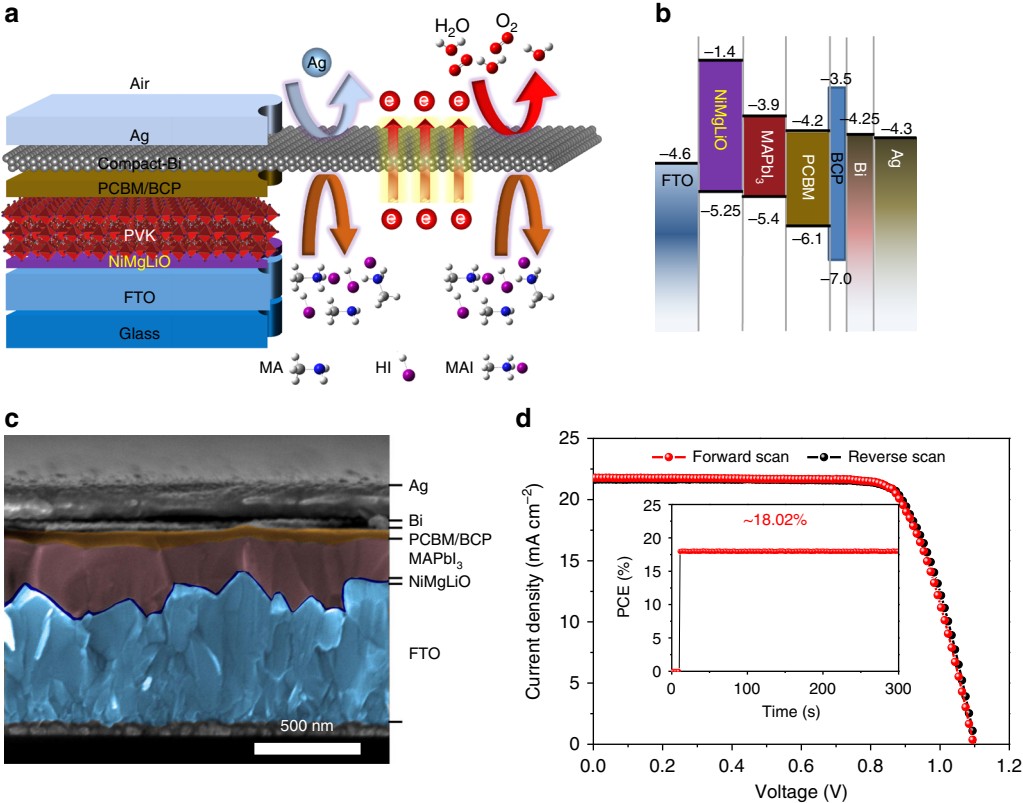

**Fig. 3** Structure, energy-level alignment, and performance of HPVKSC. **a** Schematic diagram of the device structure in this work: FTO/NiMgLiO/PVK/PCBM/BCP/Bi/Ag; the Bi interlayer has a superior shielding capability, prohibiting both inward and outward permeation. **b** Energy level diagram and **c** SEM cross-sectional image of the device (scale bar: 500 nm). **d** J–V curves of a typical large-area (1 cm²) HPVKSC with a 20-nm Bi interlayer based on a MAPbI$_3$ absorber, with the stabilized PCE measured at 0.872 V as the inset. Source data are provided as a Source Data file

curves of MA-HPVKSCs with an optimized Bi interlayer (20 nm thick). Under AM 1.5G simulated sunlight (100 mW cm$^{-2}$), the fabricated device has a PCE over 18% (a PCE of 18.01%, with an open-circuit voltage ($V_{OC}$) of 1.098 V, a short-circuit photocurrent density ($J_{SC}$) of 21.78 mA cm$^{-2}$, and a fill factor (FF) of 0.753 for the forward scan; a PCE of 18.06%, with a $V_{OC}$ of 1.102, a $J_{SC}$ of 21.62 mA cm$^{-2}$, and a FF of 0.758 for the reverse scan; stabilized PCE measured at MPP of 0.872 V is 18.02%). The PCE of Bi-based devices can be promoted to approximately 19% by changing the benchmark MAPbI$_3$ to the currently popular FA/MA/Cs triple cation perovskite ((FAMACs)Pb(IBr)$_3$) (for details, see the J–V curves and stabilized PCEs in Supplementary Figure 9 and Supplementary Figure 13, the photovoltaic parameters in Supplementary Table 5, and the performance statistics in Supplementary Figure 17 and Supplementary Figure 20).

**The impenetrability of Bi to ions.** The effect of Bi as the penetration barrier to ion diffusion has been studied. First, ToF-SIMS was used to monitor the changes in the elemental distribution for each layer before and after thermal aging at 85 °C for 100 h in the dark under N$_2$ atmosphere. As shown in Fig. 4a, b, for the controlled MA-HPVKSCs without Bi interlayers, inward diffusion of Ag$^+$ ions into the perovskite layer and outward diffusion of I$^-$ ions towards the Ag electrode layer were detected after aging. Thus, the PCBM/BCP interfacial layer is permeable for the diffusion of I$^-$ and Ag$^+$ ions. For the devices with Bi interlayers (Fig. 4c, d), the inward diffusion of the Ag$^+$ ions and outward diffusion of I$^-$ ions were effectively blocked. Thus, the inserted Bi interlayer has robust impenetrability as an ion diffusion barrier.

Second, the morphologies and photoelectronic properties of MAPbI$_3$ films in the configurations Glass/MAPbI$_3$/PCBM/BCP/Bi/Ag and Glass/MAPbI$_3$/PCBM/BCP/Ag (i.e., with and without a Bi interlayer) after thermal aging (at 85 °C for 100 h in the dark and N$_2$ atmosphere) were compared by SEM and time-resolved PL characterizations. The films were obtained by washing away the top PCBM/BCP/electrode layers with chlorobenzene. For the MAPbI$_3$ film in the configuration without Bi, the crystal grains fused, and their size increased considerably from approximately 500 nm to approximately 1100 nm (Fig. 5a, b). The large grain size of the aged MAPbI$_3$ film can be correlated with the penetrated Ag$^+$ ions (similar to the case in the study of Chen et al.[47], which found that a small amount of Ag-doping could considerably increase the resultant perovskite film's grain size.) By contrast, the morphology of the MAPbI$_3$ films in the configuration with the Bi interlayer changed only marginally after aging (Fig. 5c, d). This result indicates that the Bi interlayer is very important for maintaining the MAPbI$_3$ film's morphology; this is important because the morphology may significantly affect the electronic properties. The transient PL spectra shown in Fig. 5e indicate that with the Bi interlayer's protection, the aged-MAPbI$_3$ and aged-MAPbI$_3$/fresh-PCBM samples exhibited carrier lifetimes similar to those of fresh-MAPbI$_3$, which confirms that the electronic quality of the MAPbI$_3$ was not changed by thermal aging. In contrast, for the devices without Bi interlayers, the aged MAPbI$_3$ sample exhibited a considerably shorter carrier lifetime in comparison to the fresh one (72 ns versus 120 ns). Moreover, the MAPbI$_3$/PCBM sample exhibited deteriorated charge extraction at the interface after aging (the PL lifetime increased from 10 to 17 ns). These results confirm that the photoelectronic properties of the perovskite films degraded in the

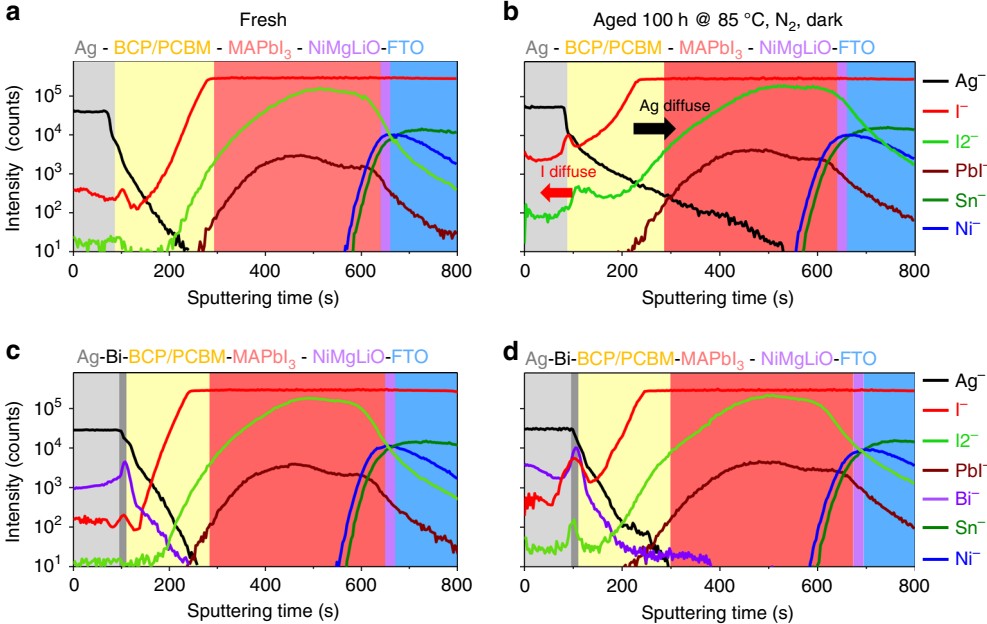

**Fig. 4** Elemental distribution in HPVKSCs. ToF-SIMS elemental depth profiles of MA-HPVKSCs. **a** and **c** are the fresh devices with and without a Bi interlayer with a thickness of 20 nm; **b** and **d** are those two devices aged at 85 °C for 100 h in the dark under $N_2$ atmosphere. Source data are provided as a Source Data file

configuration without the Bi interlayer. We also fabricated devices based on the two types of aged $MAPbI_3$ films by depositing fresh PCBM/BCP/electrodes (Fig. 5f). The $J-V$ curves are presented in Fig. 5g, h, and the corresponding stabilized PCEs were shown in Supplementary Figure 10, and the photovoltaic parameters are tabulated in Supplementary Table 4. For the device without the Bi interlayer, the thermal aging severely decreased the PCE from its initial value of 18.51% to 14.24% and increased the hysteresis (Forward Scan: aging changed $V_{OC}$ from 1.104 to 1.085 V, $J_{SC}$ from 22.06 to 20.09 mA cm$^{-2}$, FF from 0.759 to 0.670, and PCE from 18.49% to 14.61%; Reverse Scan: $V_{OC}$ from 1.107 to 1.081 V, $J_{SC}$ from 22.04 to 18.97 mA cm$^{-2}$, and FF from 0.759 to 0.676, and PCE from 18.52% to 13.86%). After renewing the PCBM/BCP/Ag layers, the PCE of the refreshed device partially recovered to 15.93% (Forward Scan: $V_{OC}$ of 1.086 V, $J_{SC}$ of 20.43 mA cm$^{-2}$, FF of 0.730, and PCE of 16.18%; Reverse Scan: $V_{OC}$ of 1.085 V, $J_{SC}$ of 19.88 mA cm$^{-2}$, FF of 0.726, and PCE of 15.67%). The recovery of PCE from 14.24% to 15.93% was likely related to the freshly deposited Ag electrode without corrosion (discussed later), whereas, the irreversible PCE loss from 18.51% to 15.93% was probably due to electronic contamination of the $MAPbI_3$ film by the inward diffusion of Ag$^+$ ions, which could not be reversed by renewing the PCBM/BCP/Ag films. In contrast, for the device with the Bi interlayer, thermal aging caused only a marginal PCE decline from 18.00% to 16.91%. The PCE decline could almost be recovered (17.50%) by renewing the PCBM/BCP/Bi/Ag layers. The results indicate that the photoelectronic properties of the $MAPbI_3$ film can be kept almost unchanged during thermal aging under the protection of the Bi interlayer, which prevents Ag diffusion into the $MAPbI_3$.

In addition to prohibiting inward diffusion of ions from the Ag to the perovskite, the Bi interlayer blocks the outward diffusion of I$^-$ ions at the inner surface of the Bi interlayer (Fig. 4c, d). We adopted SEM and XRD characterizations to investigate the changes of the inner surface of the electrodes before and after aging the devices at 85 °C for 100 h in a $N_2$ filled glovebox in the dark (see Supplementary Figure 11). The metal electrode samples

were obtained by dissolving the PCBM/BCP in chlorobenzene. For the device without the Bi interlayer, the morphology of the Ag electrode inner surface was changed significantly by thermal aging. The surface became rough with many concave pits, and the XRD patterns in Supplementary Figure 11 confirm the formation of AgI. On the contrary, the results for the devices with Bi interlayers were completely different. As shown in the SEM images (Supplementary Figure 11), the morphology of the Bi/Ag electrode inner surfaces remained almost unchanged after thermal aging. The corresponding XRD patterns confirm that no AgI or BiI$_3$ was formed. These observations clearly demonstrate the validity of the Bi interlayer's protection of the Ag electrode from corrosion by the halide perovskite.

**Stability test of the devices.** The excellent anti-corrosive property and robust impermeability of the introduced Bi interlayer may largely improve the stability of the corresponding solar cells. The stability of HPVSKSCs should be assessed under specified aging conditions, which have been well discussed in recent review papers[48–50]. A storage stability test was conducted in the dark in ambient air at room temperature (RT) without humidity control (relative humidity of the local climate is normally 40–90%). Separately, we implemented a thermal aging test at 85 °C in $N_2$ atmosphere in the dark for 500 h. 85 °C is specified for the aging temperature according to the standard IEC 61215-1:2016[51]. As proposed by Snaith et al.[49], the thermal aging atmosphere could be relaxed to an inert atmosphere ($N_2$) for eliminating the interference from other stresses from the ambient, such as $H_2O$ and $O_2$. All devices were open circuits during thermal aging. Also, we measured the operational stability of our devices at an elevated temperature of 45 °C with the external bias near their MPPs by referring to the standard IEC 61646: 2008[52]. As Grätzel et al.[50] advocated, to eliminate interference from ultraviolet light and stress from $H_2O$, and $O_2$, our light soaking test was performed in a $N_2$ filled glovebox by using a white light LED array as the light source. The light intensity was calibrated to achieve the same $J_{SC}$ from the HPVKSCs as for 1 sun AM1.5G solar irradiation.

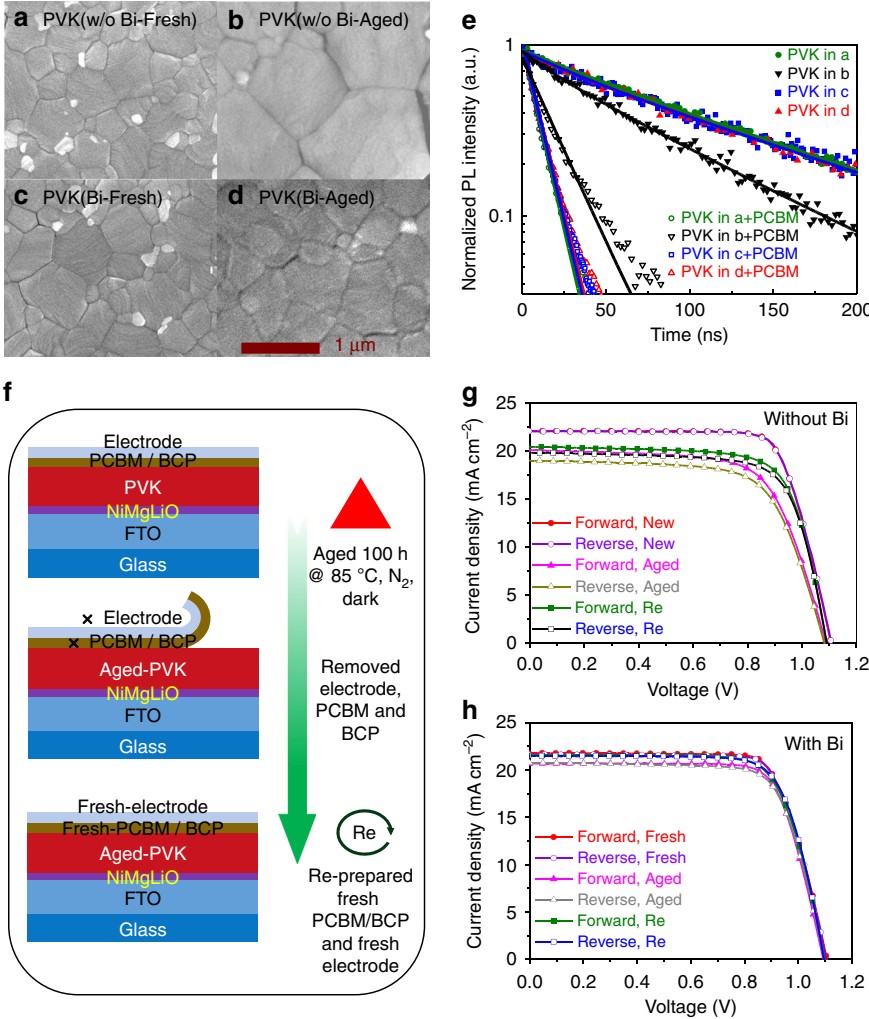

**Fig. 5** Changes of PVK layers during thermal aging. **a–d** SEM images (scale bar: 1 μm) and **e** time-resolved PL spectra of MAPbI$_3$ films in devices with the configurations Glass/MAPbI$_3$/PCBM/BCP/Bi/Ag and Glass/MAPbI$_3$/PCBM/BCP/Ag (i.e., with and without (w/o) Bi interlayers) before and after aging at 85 °C for 100 h in a N$_2$ filled glovebox in the dark. The MAPbI$_3$ films were obtained by washing away the top PCBM/BCP/electrode layers. The re-prepared devices were prepared according to the schematic diagrams shown in (**f**) by re-preparing the PCBM/BCP/electrode layers. **g**, **h** J–V curves of the HPVKSCs without (**g**) and with (**h**) Bi interlayers, based on the fresh, aged MAPbI$_3$ films, and the aged MAPbI$_3$ films with re-prepared top layers. Forward Scan: from −0.1 to 1.2 V. Reverse Scan: from 1.2 to −0.1 V. The corresponding stabilized PCEs are shown in Supplementary Figure 10. Source data are provided as a Source Data file

First, the storage stability of unencapsulated large-area (1 cm$^2$) MA-HPVKSCs was assessed. As illustrated in the normalized PCE in Fig. 6a (the performance statistics in Supplementary Figure 12, the J–V curves in Supplementary Figure 13 and their corresponding photovoltaic parameters in Supplementary Table 5), the Bi-based device maintained approximately 88% of its initial PCE after 6000 h, whereas the pristine device without the Bi interlayer maintained only approximately 33% of its initial PCE after 1000 h. The $V_{OC}$ and $J_{SC}$ of the Bi-based device were highly stable, and FF degraded slowly with the aging time. In contrast, the photovoltaic parameters of the typical devices without Bi, including $V_{OC}$, $J_{SC}$, and FF, suffered marked declines. Repeated experiments suggested that the Bi strategy can effectively improve the stability with high reproducibility (see the performance statistics in Supplementary Figure 12). To the best of our knowledge, MA-HPVKSCs using the interlayer strategy have not achieved such high stability during ambient storage (see the comparison of storage stability for selected HPVKSCs in Supplementary Table 6). Thus, the inserted Bi interlayer plays an important role in "self-encapsulation", which

can prevent the penetration of ambient moisture. The optical images of the devices with different thicknesses of the Bi interlayer after aging for 6000 h are displayed in Supplementary Figure 14. A large region of the active film in the device without Bi turned yellow, which indicates decomposition of the perovskite. The active films in the devices with the Bi interlayer (especially for a thickness greater than 20 nm) were almost unchanged. The presence of the dense Bi interlayer can prevent or delay direct exposure to external moisture and thus increase the long-term survivability of the device in the ambient air.

Second, unencapsulated large-area (1 cm$^2$) MA-HPVKSCs were investigated under thermal aging (85 °C, dark, N$_2$ atmosphere) and light soaking conditions (45 °C, white LED light, near MMP, N$_2$ atmosphere) for 500 h in a N$_2$ filled glovebox. The normalized PCEs of typical devices as a function of the aging time are shown in Fig. 6b, c. The devices with the Bi interlayer retained approximately 87% and approximately 91% of their initial PCEs after thermal aging and light soaking tests, respectively, whereas the pristine devices without Bi retained only approximately 42% and approximately 27% of their initial PCEs after the thermal

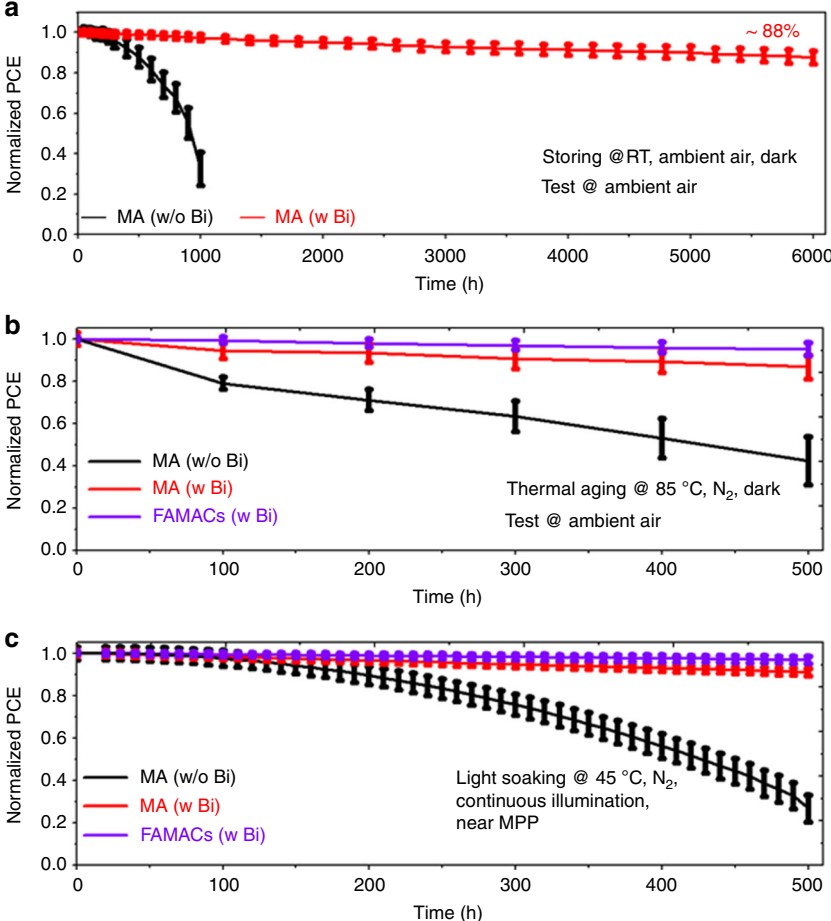

**Fig. 6** Stabilities of unencapsulated HPVKSCs. The average efficiency obtained as a function of time from 9 devices is shown with error bars that represent the standard deviation of the devices. **a** Stored in the dark in ambient air at RT without humidity control, and *J–V* curves acquired periodically in ambient air; **b** aged in the dark at 85 °C in an $N_2$ atmosphere, and *J–V* curves acquired periodically in ambient air. The device performance measurements in (**a**, **b**) were measured under standard AM1.5G simulated sunlight; **c** aged under continuous illumination in an $N_2$ atmosphere with electrical biases (0.641–0.885 V) near MPP at a cell temperature of 45 °C. The light intensity for aging was generated by a white light LED array and calibrated to achieve the same $J_{SC}$ from the HPVKSCs as for 1 sun AM1.5G solar irradiation. Source data are provided as a Source Data file

aging and light soaking tests, respectively. The Bi-based MA-HPVKSCs showed more stable $V_{OC}$, $J_{SC}$, and FF than did the devices without Bi, especially $J_{SC}$ (see the typical *J–V* curves in Supplementary Figure 13, the performance statistics of thermal stability in Supplementary Figure 15, and the performance statistics of light-soaking stability in Supplementary Figure 16). The considerably improved thermal and light-soaking stabilities were due to the robust shielding capability of the Bi interlayer.

Third, we further changed the composition of the perovskite film in our devices from the early benchmark MAPbI₃ to the currently popular FA/MA/Cs triple cation perovskite[53]. In this configuration, the Bi interlayer also had a positive effect on the devices' long-term stability. Under the same thermal aging and light soaking conditions, the (FAMACs)Pb(IBr)₃ based HPVKSCs (FAMACs-HPVKSCs) with Bi interlayers maintained approximately 95% and approximately 97% of their initial PCEs, respectively (Fig. 6b, c, see the typical *J–V* curves and stabilized PCEs in Supplementary Figure 13 and Supplementary Figure 17, and their photovoltaic parameters in Supplementary Table 5, the performance statistics of thermal stability in Supplementary Figure 18, the performance statistics of light-soaking stability in Supplementary Figure 19). By extrapolation, the time at which a typical FAMACs-HPVKSC with Bi interlayers would have degraded to 80% of its initial efficiency under light soaking was

determined to be 11,800 h. Furthermore, the statistical parameters reflect that FAMACs-HPVKSCs with Bi interlayers are highly stable and highly reproducible (see the performance statistics of thermal stability in Supplementary Figure 18, and the performance statistics of light-soaking stability in Supplementary Figure 19). The stability data of the FAMACs-HPVKSCs were superior to those of the MA-HPVKSCs and close to the international acceleration test standards for silicon solar cells (IEC 61215)[51]. Overall, our Bi interlayer strategy compares well to the recent reports regarding the improvement of long-term stability (see the comparison of storage stability for selected HPVKSCs in Supplementary Table 6, the comparison of thermal stability for selected HPVKSCs in Supplementary Table 7 and the comparison of operational stability for selected HPVKSCs in Supplementary Table 8). These results demonstrate the importance of using chemically inert barrier interlayers in the devices for achieving long-term stability and provide a promising strategy for the further application of perovskites in commercial-scale production.

## Discussion

We demonstrated Bi to be an intrinsically stable metal for halide perovskites even under harsh conditions. The XPS and XRD

analyses indicate that no reaction occurred between the Bi and perovskites including $MAPbI_3$, $(FAMA)Pb(IBr)_3$, $(FACs)Pb(IBr)_3$, and $(FAMACs)Pb(IBr)_3$ at 85 °C over an extended period. Another notable feature of the Bi film is that it can grow into compact barrier layers with scalable device sizes via a facile thermal evaporation technique. Thus, the Bi metal was introduced conveniently into inverted-structured HPVKSCs as an interlayer before the thermal-deposition of the Ag electrode. The Bi interlayer acted as a penetration barrier that insulated against the undesirable intrusion of external moisture while also suppressing the diffusion of internal ions, including the inward diffusion of $Ag^+$ ions into the perovskite layer and the outward diffusion of $I^-$ ions to the Ag electrode. As a result, the Bi-interlayer-based HPVKSCs exhibit greatly enhanced long-term stability under various humidity, thermal and light stresses. On average, the unencapsulated $MAPbI_3$-HPVKSC devices retained approximately 88% of their initial efficiency in ambient air in the dark for over 6000 h; moreover, the FAMACs-HPVKSCs maintained approximately 95% and approximately 97% of their initial efficiencies after 85 °C thermal aging and light soaking in a $N_2$ atmosphere for 500 h, respectively. These superior stability parameters are among the best for planar-structured HPVKSCs reported to date. Thanks to its low-cost, non-toxicity, and compatibility with scalable thermal evaporation processes, the Bi interlayer strategy is a commercially feasible means of resolving the problem caused by unsettled metal electrodes in practical HPVKSCs.

## Methods

**Materials**. Methylammonium iodide ($CH_3NH_3I$ (MAI), 99.99%), methylammonium bromide ($CH_3NH_3Br$ (MABr), 99.99%) and formamidinium iodide ($CH(NH_2)_2I$ (FAI), 99.99%), and formamidinium bromide ($CH(NH_2)_2Br$ (FABr), 99.99%) were purchased from Hangzhou Perov Optoelectronic Technology Co., Ltd. Lead (II) iodide (99.99% metals basis), lead (II) bromide (>98%), cesium bromide (>99%), cesium iodide (>99%), and bathocuproine (>99%) were purchased from Tokyo Chemical Industry Co., Ltd. All other chemicals were purchased from Sigma-Aldrich and used as received.

**Device fabrication**. FTO glasses (TEC-8, Nippon Sheet Glass Co., Japan) were cleaned through sequential ultrasonication for 20 min in a detergent solution, distilled water, alcohol, and acetone. Then, a p-type NiMgLiO was deposited on top of the FTO glass: a mixture solution of acetonitrile and ethanol (with 95:5 volume ratio, 30 mL) of nickel acetylacetonate (with magnesium acetate tetrahydrate and lithium acetate, and the mole atomic ratios of Ni:Mg:Li is 80:15:5, the total metal ion concentration is $0.02 \, mol \, L^{-1}$) was sprayed by an air nozzle (with 0.2 mm caliber) onto the hot FTO glasses (500 °C). After spraying, the samples were further treated at 500 °C for another 20 min and then leave them to cool naturally. After cooling to RT, perovskite was then deposited by the anti-solvent method: a 80 μL 1.5 M $MAPbI_3$ solution (a mixture of $PbI_2$:MAI = 1.05:1 by molar ratio and DMF: DMSO = 4:1 by volume) was spin-coated at 5000 rpm for 30 s, followed by quickly drop-casting diethyl ether (1000 μL) as an anti-solvent within 10 s. The $(FAMA)Pb(IBr)_3$ perovskite films were deposited by a precursor solution containing FAI (1 M), $PbI_2$ (1.1 M), MABr (0.2 M) and $PbBr_2$ (0.2 M) in DMF:DMSO = 4:1 by volume. The $(FACs)Pb(IBr)_3$ perovskite films were deposited by a precursor solution containing FAI (1.275 M), $PbI_2$ (1.52 M), and CsBr (0.225 M) in DMF: DMSO = 4:1 by volume. The $(FAMACs)Pb(IBr)_3$ perovskite films were deposited by a mixture of 1 mL $(FAMA)Pb(IBr)_3$ precursor solution and 0.05 mL CsI solution (1.5 M in DMSO). Then, the perovskite films were treated thermally at 100 °C for a period of time, 10 min for $MAPbI_3$, 30 min for $(FAMA)Pb(IBr)_3$, $(FACs)Pb(IBr)_3$, and $(FAMACs)Pb(IBr)_3$. The PCBM solution was prepared by dissolving 20 mg PCBM in 1 mL of chlorobenzene and was stirred vigorously at 45 °C overnight before spin-coating. A approximately 60-nm-thick PCBM was deposited on the perovskite/NiMgLiO/FTO substrate by spin-coating at 2000 rpm for 30 s at RT followed by a 70 °C thermal treatment for 10 min. In the final step, the BCP, Bi, and Ag were deposited at high vacuum (less than $5 \times 10^{-4}$ Pa) while finely controlling the evaporation rate at 0.1, 0.1, and $0.1–0.5 \, Å \, s^{-1}$, respectively.

**Characterization and measurement**. SEM images were obtained using a Nova NanoSEM 450 scanning electron microscope (FEI Co., Netherlands). An XPS system (Thermo ESCALAB 250XI) was used to acquire the XPS spectra. The crystal structure of the films was characterized using XRD with an Empyrean X-ray diffractometer with Cu Kα radiation (PANalytical B.V. Co., Netherlands). ToF-SIMS depth profiles were carried out using an IonTof ToF-SIMS 5 instrument (IONTOF Co., Germany). The work functions of the samples were investigated using a KP Technology Ambient Kelvin probe system package. The time-resolved PL studies were performed using an Edinburgh FLS920 fluorescence spectrometer (Edinburgh Co., UK). The UV–vis spectra were recorded using a Lambda 950 spectrophotometer (PerkinElmer Co., USA). Photovoltaic measurements employed a black mask with an aperture area of 1 $cm^2$ under standard AM1.5G simulated sunlight (Oriel, model 9119) and the simulated light intensity was calibrated with a silicon photodiode. Incident photon-to-electron conversion efficiency (IPCE) was measured using a Newport IPCE system (Newport, USA). The light-soaking stability was measured with a CHI1000c multichannel electrochemistry workstation (Chenhua, Co., China).

## Data availability

The data that support the findings of this study are available from the corresponding author upon request.

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

## Acknowledgements

This work was financially supported by the National Natural Science Foundation of China (51672094, 51861145404, and 51822203), the China Postdoctoral Science Foundation Grant (2016M602286), the Self-determined and Innovative Research Funds of HUST (2016JCTD111), Shenzhen Science and Technology Innovation Committee (JCYJ20170307165905513), and Natural Science Foundation of Guangdong Province (2017A030313342). The authors thank Analytical and Testing Center of Huazhong University Science and Technology for the sample measurements.

## Author contributions

S.W., R.C., S.Z. and Wei Chen conceived the project and designed the experiments. S.W., R.C. and S.Z. performed and were involved in all the experimental parts. S.W., R.C., S.Z., B.H.B., Y.Y., Wei Chen, and L.H. co-wrote the paper. Weitao Chen, C.C., Z.Y., Y.H., H.Z., S.F. and T.L. contributed materials and analysis tools. L.H. participated in the supervision of the work. Wei Chen directed and supervised this project. All authors discussed the results and commented on the manuscript.

## Additional information

**Competing interests:** The authors declare no competing interests.

