## [Peer Review File · Nature Communications]

Reviewers' comments:

Reviewer #1 (Remarks to the Author):

Thin and compact Bi interlayers in inverted perovskite solar cells are reported. Specifically, a tri-layer of PCBM/BCP/Bi is used to improve stability. The Bi layer is aimed to function as permeation barrier against moisture and halide molecules. Moreover, the low reactivity of Bi with halides is exploited to render the device more resilient. The overall improved stability of the PSCs is good, and the paper is of interest. The approach warrants publication in this journal. However, some major revisions are mandatory:

- 1) The claim that ALD is complicated is not entirely true, as recently even roll-to-roll compatible spatial ALD has been shown to afford suitable barrier layers (e.g. ACS Applied Mater. & Interfaces 10, 6006 (2018))
- 2) The authors showed shelf life stability in ambient air, at elevated temperature (85°C), or upon light soaking (45°C). But, to show real stability, multi-stress conditions are required, e.g. heat and illumination (AM1.5 full) with the cell kept in the maximum power point.
- 3) ToF-SIMS: Why do they see an increase of the Sn trace (from the FTO) already within the perovskite? Is this due to the roughness of the FTO? Why is the FTO so unusually rough?
- 4) For the work function they mention a value and state "versus vacuum". Note, WF is always the energetic difference of Fermi-Level and Vacuum level. So, stating a WF versus vacuum is redundant and suggests that there would be another WF defined in a different way.
- 5) It is hard to believe that the semimetal Bi causes a notable series resistance when the thickness is increased to 40 nm, especially as 60 nm PCBM are there. The conductivity of PCBM is certainly much less than that of Bi. Please explain!
- 6) Device statistics are required for all experiments (fresh and aged). How many cells have been measured? Average values and standard deviation need to be provided in order to exclude cherry-picking. Moreover, for all devices hysteresis need to be provided. Importantly, all devices need to be assessed in their maximum power point (MPP tracking) and stabilized values (efficiency, etc.) need to be provided.
- 7) Where does the energy line up in Fig 2 come from? Did the authors determine the numbers by themselves, or are these figures from the literature? In the latter case references are required.
- 8) The language needs some careful attention by a native speaker. This is required to improve readability and to avoid misconceptions!

Reviewer #2 (Remarks to the Author):

The manuscript by Wu et al. reports an interesting device engineering method that utilizes thermal evaporated Bi interface for stable perovskite solar cells.

It shows good engineering and interesting progress with respect to the device processing with simple evaporation, conductive, and conformal protection layer in perovskite solar cells.

However, the progress is incremental at best. In fact the performance parameters (1 cm² PCE of 18% and FF of 76% etc...) are not comparable to state-of-art perovskite solar cells that use same "p-i-n" structure (1. NATURE ENERGY, DOI: 10.1038/nenergy.2017.102; 2. NATURE ENERGY, DOI: 10.1038/s41560-018-0219-8; 3. Science, DOI: 10.1126/science.aap9282). Overall, I cannot recommend publishing it in Nature Communications. Some comments further listed below.

- 1). The degradation of MAPbI₃-based perovskite is widely reported in literatures. Although the Bi may

impede possible degradation path to Ag electrode by blocking I⁻ and H₂O, the internal degradation of perovskite itself cannot be avoided in such a way. The instability of MAPbI₃ is reported by various works that show film degradation because of degassing MA (1. B. Conings et al., *Adv. Energy Mater.* 5, 1500477 (2015); 2. W. Tan, A. R. Bowring, A. C. Meng, M. D. McGehee, P. C. McIntyre, *ACS Appl. Mater. Interfaces* 10, 5485–5491 (2018); 3. S. H. Wang, Y. Jiang, E. J. Juarez-Perez, L. K. Ono, Y. B. Qi, *Nat. Energy* 2, 16195 (2017); 4. T. Zhang et al., *J. Mater. Chem. A Mater. Energy Sustain.* 5, 1103–1111 (2017); 5. Y. C. Zhao et al., *J. Phys. Chem. C* 121, 14517–14523 (2017). The volatile nature of the MA molecule itself using the degradation pathway of CH₃NH₃I into CH₃I and NH₃ at temperatures as low as 80°C. Please double check the 85°C stability data.

2). The 85°C thermal stability test conditions should be specified here, such as open circuit, short circuit or V_{mpp} conditions.

3). The conductivity of Bi is not comparable to FTO, otherwise, your Bi thickness would not be limited at 20 nm. The enhancement of R_s and drop of FF is obvious when the thickness goes to 40 nm. In fact, the conductivity of Bi layer one order of magnitude lower than regular FTO.

Reviewer #3 (Remarks to the Author):

Review of Wu et al. "A chemically inert bismuth interlayer enhances long-term stability of inverted perovskite solar cells"

The work by Wu et al. presents a quite compelling device study on the use of a bismuth metallic interlayer in an inverted perovskite photovoltaic device. Although much of the data is of considerable interest the insight provide by the work is somewhat limited and precludes the reviewer from recommending it for publication in Nature communications. The use of the Bi metal is somewhat novel bulk of the work carried out on MAPbI₃ with targeted demonstrations of mix cations appears well conceived and interesting so if the following issues are addressed comprehensively it might become suitable for publication.

Specifically the authors fail to cite the relevant ISOS protocols used for the device stability testing or reference the relevant protocols (doi: 10.1016/j.solmat.2011.01.036). More serious is the providing of a rational for the test protocols that are employed. The rational for shelf life test vs the 85°C N₂ testing in the dark in comparison to 45°C N₂ testing under continuous illumination is not provided by the authors. While it is clear that the inclusion of Bi improves the devices the article provides little insight as to why this strategy is succesfull or indeed why it maybe succesfull and thus the impact of the work is significantly reduced as how this might be translated by other researchers is unclear. This is exemplified by the lack of a clear rational for the testing mentioned above and although there maybe a reasonable rational for these test it is not provided within the manuscript. This issues must be addressed given the results by Domanski et al. (<https://doi.org/10.1038/s41560-017-0060-5>) which should be referenced. Data presented in the SI are in many ways more compelling as it suggest a more general set of design rules and a overarching framework for the choice of metal interlayers which might be less susceptible to chemical instabilities within the device stack.

It is notable that other important work in stability is also not referenced particularly work by NREL (DOI: 10.1021/acsenerylett.6b00013,<https://doi.org/10.1038/s41560-017-0067-y>) or on the chemistry of the fullerene based interlayers used in many composite by researchers at the Univ of Washington (notably DOI: 10.1002/adma.201300580) which indicates the types of chemistries likely planing an important role at these functional device interfaces as well as this groups more recent work

directly in perovskite solar cells. Similarly the question of the metallic interfaces analogs and associated oxide literature is also not well referenced (DOI: 10.1021/acsami.6b10898 and DOI: 10.1021/acscenergylett.6b00320) with the later appearing to be directly relevant to the work reported here and providing a framework that could rationalize the material choice and its stability performance. The issue of metal ingress generally has also been addressed by McGehee group with recent work on stable cell architectures and packages, the referee would suggest the implicates of the presented work in the context of these recent publications would increase the impact of this work.

The above issues are the most critical to be addressed as if done so can address the confusion that currently exists within the paper regarding what the Bi layer is actually doing. The current confusion is result of the authors not providing a clear picture or posing a clear hypothesis of what is actually happening. At present multiple hypotheses appear but without any really being tested comprehensively leaving the reader with simply the device data and that the Bi based systems appear to be a bit better.

It appears to this reviewer that in the context of the above literature the existing data can be better understood and this understanding presented given with the exiting data. Specifically the criteria of anti-corrosive metals and the associated deposition conditions should be expanded on. Related to this is the ambiguous use of the description of "anti-corrosive" evaporation and metal systems in the context of this device work. It is the referees perspective that the associated discussion is perhaps one of the most compelling aspects of the work, corrosion appears to be an accurate general descriptor but should the existing draft does not provide enough specificity regarding the actual corrosion process/chemistry in this case. While the reviewer understands the choice of examining the PCBM/BCP/Bi/Ag and related layers it would be instructive if to examine the MAPbI₃/Bi layers and associated material morphology, structure and interfacial chemistry to understand the potential advantages of the Bi interface.

Other minor issues which should also be addressed. As an example perovskite solar cells (PVKSC) is probably not a sufficiently specific designation (nor is the other common designation of PSC currently often used for perovskite solar cells as it is also commonly used for plastic solar cells) as this can also refer to the oxide systems which have been demonstrated but operate on bulk photovoltaic effect and leverage ferroelectric behaviors. The existing work would seem to apply to any of the halide perovskite solar cell systems and some designation reflecting this should be used (i.e. HPVSKSC or HPSC for halide perovskite solar cells systems which the work targets). Similarly the data in figure 3 (TOF-SIMS) appears to have the halide signal (I-) completely saturated and no clear indication of the polarity (while this can be inferred from the data). Although these issues with the TOF-SIMS doesn't undercut the insight from these measurements these details should be provided in a more complete way (see recent review on this technique for halide perovskite solar cells DOI:10.1021/acsami.8b07937).

A point-by-point respond to reviewer's comments

Reviewer #1:

Thin and compact Bi interlayers in inverted perovskite solar cells are reported. Specifically, a tri-layer of PCBM/BCP/Bi is used to improve stability. The Bi layer is aimed to function as permeation barrier against moisture and halide molecules. Moreover, the low reactivity of Bi with halides is exploited to render the device more resilient. The overall improved stability of the PSCs is good, and the paper is of interest. The approach warrants publication in this journal. However, some major revisions are mandatory:

Q1: The claim that ALD is complicated is not entirely true, as recently even roll-to-roll compatible spatial ALD has been shown to afford suitable barrier layers (e.g. ACS Applied Mater. & Interfaces 10, 6006 (2018))

Responses: Thanks for the comments. We have removed the inappropriate description in the revised manuscript. Previous description “However, the technical process of ALD was relatively complicated because it involves layer-by-layer hydrolysis and condensation reactions, which may cause problems such as declined efficiency or increased processing cost. Thus, a simple and reproducible technology for the preparation of non-permeable interlayers with robust chemical stability is highly desired.” has been removed.

Q2: The authors showed shelf life stability in ambient air, at elevated temperature (85°C), or upon light soaking (45°C). But, to show real stability, multi-stress conditions are required, e.g. heat and illumination (AM1.5 full) with the cell kept in the maximum power point.

Responses: Thanks for the reviewer's concern on this point. Actually, we did our accelerated tests according to the protocols which have been recommended in recent two review papers published on “**Nature Energy**”, and also by referring to the international standards (EC 61646:2008 and IEC 61215-1:2016). And we have supplemented appropriate description in the revised manuscript (see the part of “**Stability test of the devices**”, page 18, line 4 to page 19, line 1, highlighted by the yellow color). See below are the screenshot of these two references.

[Redacted]

As suggested by literature 1 (Domanski, K., Alharbi, E. A., Hagfeldt, A., Grätzel, M. & Tress, W. Systematic investigation of the impact of operation conditions on the degradation behaviour of perovskite solar cells. *Nat. Energy* **3**, 61–67 (2018)), the aging conditions of light soaking for operational stability are: 1 sun solar simulator, possible to relax to white LEDs (in our work: white LED light), MPP (in our work: MPP), room and elevated temperatures (in our work: 45 °C), inert environment (in our work: N₂ filled glovebox).

As suggested by literature 2 (Snaith, H. J., & Hacke, P. Enabling reliability assessments of pre-commercial perovskite photovoltaics with lessons learned from industrial standards. *Nat. Energy* **3**, 459-465 (2018)), the aging conditions for thermal stability are: dark (in our work: dark), 60, 85 °C or higher (in our work: 85 °C) and inert, dry air, ambient air with quoted RH, 85% RH (in our work: N₂ filled glovebox).

See above, the aging conditions used in our work almost match well the suggested ones from the two literatures. Two points need to discuss: (1) A N₂ atmosphere was employed here instead of more challenging conditions of ambient air or air with high humidity in order to avoid the uncontrolled impacts from insufficient encapsulation. But this atmosphere is popular in recent PSCs literatures, because perfect encapsulation is only a technical issue; (2) 45 °C was used here, which is a little bit lower than 60 °C for light soaking stability tests in many PSCs literatures, however, if referring to the IEC 61646:2008 (Thin-film Terrestrial Photovoltaic Modules—Design, Qualification and Type Approval), they clearly defined that the device temperature of light soaking could be 50 °C ± 10 °C. Therefore, we think the device temperature of 45 °C used in our work is acceptable for light soaking tests.

Q3: ToF-SIMS: Why do they see an increase of the Sn trace (from the FTO) already within the perovskite? Is this due to the roughness of the FTO? Why is the FTO so unusually rough?

Responses: Thanks for the reviewer’s concern on this point. We think the Sn trace should be caused by high roughness of the FTO glasses (NSG, TEC-8, from Nippon Sheet Glass Co., Japan) used here. As shown in the following cross-sectional SEM image (left, also see in the **Figure 3c**), the roughness of the FTO could be as high as 200 nm. The following table (right) was provided by NSG Co., Japan. The haze of TEC-8 is much higher than other kinds FTO glasses. Normally, such high haze FTO glass is preferably used in dye-sensitized solar cells (Chiba, Y. *et al.* Dye-sensitized solar cells with conversion efficiency of 11.1%. *Japan. J. Appl. Phys.* 2 **45**, L638–L640 (2006)), which is beneficial for increasing the optical path length to enhance light-harvesting efficiency. It was selected to use in our PSCs because of similar reason.

Q4: For the work function they mention a value and state “versus vacuum”. Note, WF is always the energetic difference of Fermi-Level and Vacuum level. So, stating a WF versus vacuum is redundant and suggests that there would be another WF defined in a different way.

Responses: Thanks for the comments. We have corrected the description about the work function. Please see page 13, line 6-9, highlighted by the yellow color.

Q5: It is hard to believe that the semimetal Bi causes a notable series resistance when the thickness is increased to 40 nm, especially as 60 nm PCBM are there. The conductivity of PCBM is certainly much less than that of Bi. Please explain!

Responses: Thanks for reminding us to deeply think about this point. Actually, we previously have similar confusion. This time, we have carefully checked the Bi film’s crystal growth mechanism, and we have found the inherent reason leading to the abnormally increased series resistance.

As shown in the SEM and AFM images below (as well as in the revised **Supplementary Figure S8**), when the thickness of the Bi films increases from 10 nm,

40 nm to 80 nm, the surface roughness changes significantly from 3.7 nm, 12.3 nm to 37.7 nm. Some tower-shaped protuberances with the height of ~ 300 nm could be detected in the 80 nm-thick Bi film. We think the high roughness of Bi film will lead to an increased contact resistance between Bi and Ag, and therefore a notably increased series resistance of the solar cell (similar phenomenon has been reported in the reference: Schubert, S., Meiss, J., Müller-Meskamp, L., & Leo, K.. Improvement of transparent metal top electrodes for organic solar cells by introducing a high surface energy seed layer. *Adv. Energy Mater.* **3**, 201200903 (2013)). Related description has been added in the revised manuscript. Please see page 13, line 20 to page 14, line 4, highlighted by the yellow color.

Supplementary Figure S8. AFM (scale bar: 1 μm) and SEM (scale bar: 500 nm) cross-sectional images of the Bi layers (onto the $\text{MAPbI}_3/\text{PCBM}/\text{BCP}$) with the thickness of 10 nm (**a, d**), 40 nm (**b, e**), and 80 nm (**c, f**).

Q6: Device statistics are required for all experiments (fresh and aged). How many cells have been measured? Average values and standard deviation need to be provided in order to exclude cherry-picking. Moreover, for all devices hysteresis need to be provided. Importantly, all devices need to be assessed in their maximum power point (MPP tracking) and stabilized values (efficiency, etc.) need to be provided.

Responses: Thanks for the suggestions. Device statistics for all experiments have been added (See **Supplementary Figure S12**, **Supplementary Figure S15**, **Supplementary Figure S16**, **Supplementary Figure S18** and **Supplementary Figure S19**). The device number have been clearly stated in the revised manuscript

(see the revised titles of **Figure 6**). The required devices hysteresis has been provided in the revised manuscript (see **Supplementary Figure S9**, **Supplementary Figure S12**, **Supplementary Figure S15**, **Supplementary Figure S16**, **Supplementary Figure S18** and **Supplementary Figure S19**). MPP tracking conditions and stabilized *PCEs* have been also complemented (see **Supplementary Figure S9**, **Supplementary Figure S10** and **Supplementary Figure S17**).

Q7: Where does the energy line up in Fig 2 come from? Did the authors determine the numbers by themselves, or are these figures from the literature? In the latter case references are required.

Responses: Thanks for your kind reminding. The energy levels in Fig 2 (in reversed manuscript **Figure 3**) are referred to literatures, which have been added in the revised manuscript. Please see page 12, line 16-17, highlighted by the yellow color.

Q8: The language needs some careful attention by a native speaker. This is required to improve readability and to avoid misconceptions!

Responses: Thanks for the comments. The language has been revised carefully by a native English speaker.

Reviewer #2:

The manuscript by Wu et al. reports an interesting device engineering method that utilizes thermal evaporated Bi interface for stable perovskite solar cells.

It shows good engineering and interesting progress with respect to the device processing with simple evaporation, conductive, and conformal protection layer in perovskite solar cells.

Q1: However, the progress is incremental at best. In fact the performance parameters (1 cm² PCE of 18% and FF of 76% etc...) are not comparable to state-of-art perovskite solar cells that use same “p-i-n” structure (1. NATURE ENERGY, DOI: 10.1038/nenergy.2017.102; 2.NATURE ENERGY, DOI: 10.1038/s41560-018-0219-8; 3. Science, DOI: 10.1126/science.aap9282). Overall, I cannot recommend publishing it in Nature Communications. Some comments further listed below.

Responses: Thanks for the comments. However, we cannot agree with the reviewer on this point based on the following reasons.

- (1) First of all, besides the efficiency, lifetime (or stability) and cost, i.e., the golden triangle, are considered to gauge the technical feasibility for commercialization of photovoltaic technologies (Meng, L., You, J. & Yang, Y. Addressing the stability issue of perovskite solar cells for commercial applications. *Nat. Commun.* **10**, 5265 (2018)). The primary task of this work is to resolve the stability issue, which is even more critical than improving efficiency in the halide perovskite solar cells (HPVKSCs) research. In all of the literatures mentioned by the reviewer (1. NATURE ENERGY, DOI: 10.1038/nenergy.2017.102; 2. NATURE ENERGY, DOI: 10.1038/s41560-018-0219-8; 3. Science, DOI: 10.1126/science.aap9282), Cu electrodes were used, which is original from Huang Jinsong’s paper (Zhao, J. *et al.* Is Cu a stable electrode material in hybrid perovskite solar cells for a 30-year lifetime? *Energy Environ. Sci.* **12**, 3650-3656 (2016)). But Cu is chemically reactive with halides perovskite, even in Huang’s paper, the by-product of CuI due to corrosion at 90 °C in the N₂ atmosphere has been reported. In Zhang’s calculation paper (Ming, W., Yang, D., Li, T., Zhang, L. & Du, M.-H. Formation and diffusion of metal impurities in perovskite solar cell material CH₃NH₃PbI₃: implications on solar cell degradation and choice of electrode. *Adv. Sci.* **5**, 1700662 (2018)), Cu is not a good choice as electrode in the halide perovskites-based devices. See also in many other literatures (Boyd, C. C. *et al.* Barrier design to prevent metal-induced degradation and improve thermal stability in perovskite solar cells. *ACS Energy Lett.* **3**, 1772-1778 (2018).), Cu could be easily to deteriorate halides perovskites when directly contacted with them even in inert atmosphere. This corrosion tendency is confirmed by our added XPS and XRD experiments (see the revised **Figure 1**, **Supplementary Figure S1**,

Supplementary Figure S2 and **Supplementary Figure S3**). Though, in the reviewer mentioned literature of “3. Science, DOI: 10.1126/science.aap9282”, Cu-based PSCs has also been reported with good stability (Their optimized devices showed a slight decay (~5%) in *PCE* after aging for 500 hours at 85°C), however, the improved stability may be arisen from other improved factors (not only from Cu electrode), such as a secondary growth perovskite layer used in their work. In the contrast, our XPS and XRD data confirm that there is no corrosion reaction between halides perovskites and Bi even under harsh conditions (see **Figure 1**, **Supplementary Figure S1**, **Supplementary Figure S2** and **Supplementary Figure S3**). In this study, the chemically inert Bi interlayer was introduced in HPVKSCs for the first time, and has achieved solid effect on stability improvement of the devices. See the summaries on stability parameters for selected HPVKSCs in **Supplementary Table S6**, **S7** and **S8**, our devices’ stability is among the best for planar structured HPVKSCs reported to date.

Supplementary Table S6. Comparison of the storage stability for selected HPVKSCs.

Device structure	Area (cm ²); Initial PCE (%); Encapsulation	T (°C); Humidity; Atmosphere	Stability (PCE / PCE _{initial})	Ref.
TiO ₂ /(CsFAMA)Pb(IBr) ₃ /HSPHCN/spiro-OMeTAD/Au	1 cm ² ; 19.6 %; No	20°C; n.a; Ambient air	~80% after 6336 h	Cheng, Y.-B. et al. 2018. ⁸
NiMgLiO/MAPbI ₃ /G-PCBM/CQDs/Ag	1 cm ² ; 17.4 %; Yes	22°C; n.a; Ambient air	~86% after 5000 h	Han, L. et al. 2017. ⁹
PEDOT:PSS/(BA) ₂ (MA) ₃ Pb ₄ I ₁₃ /PCBM/Al	1 cm ² ; 11.6 %; Yes	~25°C; 65 %RH; humidity chamber	>80% after 2250 h	Mohite, A. D. et al. 2016. ¹⁰
PEDOT:PSS/MAPbI ₃ /PCBM/AZO/SnO _x /Ag	0.018 cm ² ; 11 %; No	23°C; 50 %RH; Ambient air	~100% after 350 h	Riedl, T. et al. 2017. ¹¹
SnO ₂ /(CsFAMA)Pb(IBr) ₃ /spiro-OMeTAD/PBDB-T/Au	0.09 cm ² ; n.a.; No	20~25°C; 25 %RH; Ambient air	>90% after 3900 h	Li, G. et al. 2018. ¹²
NiMgLiO/MAPbI ₃ /PCBM/BCP/Bi/Ag	1 cm ² ; 18.05 %; No	20~30°C 40-90 %RH; Ambient air	>88% after 6000 h	This work

Supplementary Table S7. Comparison of the thermal stability for selected HPVKSCs.

Device structure	Area (cm ²); Initial PCE (%); Encapsulation	T (°C); Humidity; Atmosphere	Stability (PCE/PCE _{initial})	Ref.
NiMgLiO/MAPbI ₃ / G-PCBM/CQDs/Ag	1 cm ² ; 15.3 %; Yes	85°C; 50 %RH; Ambient air	>98% after 500 h	Han, L et al. 2017. ⁹
TiO ₂ /(CsFAMA)Pb(IBr) ₃ / HSPbCN/spiro-OMeTAD/Au	1 cm ² ; 17.5 %; Yes	60°C; n.a.; Ambient air	>70% after 384 h	Cheng, Y.-B. et al. 2018. ⁸
PEDOT:PSS/MAPbI ₃ /PCBM/AZO/SnO ₂ /Ag	0.018 cm ² ; 12 %; No	60°C 0 %RH; N ₂	~100% after 1000 h	Riedl, T. et al. 2017. ¹¹
NiO/(FACs)Pb(IBr) ₃ /LiF/PCBM/SnO ₂ /ZTO/ITO/LiF/Ag	0.12 cm ² ; 10 %; Yes	85°C; 85 %RH; humidity chamber	~100% after 1000 h	McGehee, M.D. et al. 2017. ¹³
NiMgLiO/MAPbI ₃ / PCBM/BCP/Bi/Ag	1 cm ² ; ~18.0 %; No	85°C 0 %RH; N ₂	>86% after 500 h	This work
NiMgLiO/ (FAMACs)Pb(IBr) ₃ / PCBM/BCP/Bi/Ag	1 cm ² ; ~18.7 %; No	85°C 0 %RH; N ₂	>95% after 500 h	This work

Supplementary Table S8. Comparison of the operational stability for selected HPVKSCs.

Device structure	Area (cm ²); Initial PCE (%);	Aging Temperature and Illumination	Tracking Time (h)	PCE Change in Dark (%)	Estimated T _{SS0} ^a (h)	Ref.
c-TiO ₂ /TiO ₂ / (RbFAMACs)Pb(IBr) ₃ / spiro-OMeTAD/Au	1 cm ² ; 17 %;	85°C, white LED, 100 mW cm ⁻²	500 h	+5%	2,000 h	Saliba, M. et al. 2016. ¹⁴
Cl-TiO ₂ / (FAMACs)Pb(IBr) ₃ / spiro-OMeTAD/Au	1.1 cm ² ; 20 %;	room temperature, full solar spectrum, 100 mW cm ⁻²	500 h	+10%	3,000 h	Tan, H. et al. 2017. ¹⁵
PTAA/ (FAMACs)Pb(IBr) ₃ / PS/C60/Cu	1 cm ² ; 16 %;	room temperature, white LED, 100 mW cm ⁻²	170 h	n.a.	9,000 h	Stolterfoht, M. et al. 2017. ¹⁶
NiO/(FACs)Pb(IBr) ₃ /LiF/PCBM/SnO ₂ /ZTO/ITO/LiF/Ag	0.12 cm ² ; 13 %	35°C, full solar spectrum, 100 mW cm ⁻²	1000 h	n.a.	∞	McGehee, M.D. et al. 2017. ¹³
NiO/(CsMA)PbI ₃ /PCBM/BCP/AZO/Ag/Al ₂ O ₃	0.1 cm ² ; 16.5 %	85°C, 1 Sun illumination, 100 mW cm ⁻²	1000 h	n.a.	1,500h	Park, N.- G. et al. 2018. ¹⁷
NiMgLiO/ MAPbI ₃ / PCBM/BCP/Bi/Ag	1 cm ² ; 17.75%	45°C, white LED, I _b ^b	500 h	n.a.	2,000 h	This work
NiMgLiO/ (FAMACs)Pb(IBr) ₃ / PCBM/BCP/Bi/Ag	1 cm ² ; 18.72%	45°C, white LED, I _b ^b	500 h	n.a.	11,800 h	This work

(2) Second, though the efficiency is not the primary task of this work, the performances of our devices are also competitive to the best inverted structured HPVKSCs reported to date. See the summary of the best inverted structured HPVKSCs in the following Table. Our best devices achieved PCEs of ~19%, which is very close to the highest data (19.19%) for large area PSCs based on FTO substrate (Wu, Y. et al. Thermally stable MAPbI₃ perovskite solar cells with efficiency of 19.19% and area over 1 cm² achieved by additive engineering. *Adv. Mater.* **29**,

170173 (2017)). Other high efficiency PSCs are based on ITO glasses, which is advantageous for its smaller sheet resistance and therefore higher FF of the device (Zhang, M., et al. Electrode Design to Overcome Substrate Transparency Limitations for Highly Efficient 1 cm^2 Mesoscopic Perovskite Solar Cells. *Joule* **12**, 2694-2705 (2018)). Furthermore, there is still a little room for efficiency improvement, especially for the relatively poor V_{OC} of our devices. Compared with the best inverted structured HPVKSCs reported to date, our device shows a relatively low V_{OC} , which may be further improved in the future by the interface passivation or gradient perovskites as mentioned in the literatures. However, the central theme of our work is to improve the stability rather than efficiency.

No.	Structure	Aera (cm^2)	Voc (V)	Jsc (mA/cm)	FF	PCE (%)	Strategy	Ref.
1	ITO/PTAA/(FAMA)Pb(IBr) ₃ /Passivation/C60/BCP/Cu	0.0716	1.131	22.99	79.1	20.69	interface passivation	DOI: 10.1038/nenergy.2017.102
2	ITO/PTAA/PFN-P2/(FAMACs)Pb(IBr) ₃ /LiF/C60/BCP/Cu	1.013	1.145	22.37	77.61	19.63	interface passivation	DOI: 10.1038/s41560-018-0219-8
3	ITO/PTAA/(FAMA)Pb(IBr) ₃ /PCBM/C60/BCP/Cu	0.05216	1.175	21.86	81.37	20.9	gradient perovskite	DOI: 10.1126/science.aap9282
4	FTO/NiMgLiO/MAPbI ₃ /PCBM/BCP/Ag	1.025	1.122	23.17	75.7	19.19	Perovskite quality	Wu, Y. et al. Adv. Mater. 29 , 170173 (2017).
5	FTO/NiMgLiO/(FAMACs)Pb(IBr) ₃ /PCBM/BCP/Ag	1.02	1.103	22.61	0.768	19.15	/	This work
6	FTO/NiMgLiO/(FAMACs)Pb(IBr) ₃ /PCBM/BCP/Bi/Ag	1.02	1.102	22.54	0.751	18.67		This work

(3) Third, the cost of TCO (transparent conducting oxide) was over 50% of the total cost in the perovskite module (Cai, M. *et al.* Cost-performance analysis of perovskite solar modules. *Adv. Sci.* **4**, 1600269 (2016)). Thus, it is suggested that the devices based on FTO glass/NiO hole transport layer used in our work should be more economic competitive (especially important for mass production in the industry) than ITO glasses/PTAA used in the literatures. That is because ITO glass based on rare element of Indium should be more expensive than FTO glass; besides, the organic semiconductor PTAA is also more expensive than inorganic NiO based HTL.

Q2: The degradation of MAPbI₃-based perovskite is widely reported in literatures. Although the Bi may impede possible degradation path to Ag electrode by blocking I- and H₂O, the internal degradation of perovskite itself cannot be avoided in such a way. The instability of MAPbI₃ is reported by various works that show film degradation because of degassing MA (1. B. Conings *et al.*, *Adv. Energy Mater.* **5**, 1500477 (2015); 2. W. Tan, A. R. Bowring, A. C. Meng, M. D. McGehee, P. C. McIntyre, *ACS Appl. Mater. Interfaces* **10**, 5485–5491 (2018); 3. S. H. Wang, Y. Jiang, E. J. Juarez-Perez, L. K. Ono, Y. B. Qi, *Nat. Energy* **2**, 16195 (2017); 4. T. Zhang *et al.*, *J. Mater. Chem. A Mater. Energy Sustain.* **5**, 1103–1111 (2017); 5. Y. C. Zhao *et al.*, *J. Phys. Chem. C* **121**, 14517–14523 (2017). The volatile nature of the MA molecule itself using the degradation pathway of CH₃NH₃I into CH₃I and NH₃ at temperatures as low as 80°C. Please double check the 85°C stability data.

Responses: Thanks for the comments. The reviewer 2 probably thinks that the MAPbI₃ based PSCs can't be stable under 85 °C thermal aging. Because many literatures have reported that the MAPbI₃ is easier to decompose at elevated temperature. But indeed, the degradation can be suppressed. Many recent literatures reported by independent reputable researchers have shown that the 85 °C thermal stability of MAPbI₃ based HPVKSCs (MA-HPVKSCs) could be largely enhanced, if special barrier design is implemented. For example, McGehee *et al.* even found that their MA-HPVKSCs did not decay after 1000 hours at 85 °C in the dark when an ALD-deposited SnO₂ barrier with a chemically inert ITO electrode was implemented (Boyd, C. C. *et al.* Barrier design to prevent metal-induced degradation and improve thermal stability in perovskite solar cells. *ACS Energy Lett.* **3**, 1772-1778 (2018)). Park *et al.* reported that the MA_{0.95}Cs_{0.05}PbI₃ based HPVKSCs with ALD-deposited AZO barrier layer could maintain 86.7% of the initial efficiency after 500 hours of light aging at 85 °C in ambient air (Seo, S., Jeong, S., Bae, C., Park, N.-G. & Shin, H. Perovskite solar cells with inorganic electron and hole transport layers exhibiting long-term (≈ 500 h) stability at 85°C under continuous 1 sun illumination in ambient air. *Adv. Mater.* **30**, 1801010 (2018)). Han *et al.* reported that their

MA-HPVKSCs with nano-carbon based barrier layer could maintain over 98% of initial efficiency after aging at 85 °C for 500 hours (Bi, E. *et al.* Diffusion engineering of ions and charge carriers for stable efficient perovskite solar cells. *Nat. Commun.* **8**, 15330 (2017)). Kim *et al.*'s work reported that the appearance and absorption characteristics of packaged MAPbI₃ films were almost unchanged before and after aging at 85 °C/85% RH for 1000 hours (Kim, N.-K. *et al.* Investigation of thermally induced degradation in CH₃NH₃PbI₃ perovskite solar cells using in-situ synchrotron radiation analysis. *Sci. Rep.* **7**, 4645 (2017)).

In our work, we have carefully confirmed the stability of Bi-HPVKSCs for many times before writing this paper. And the stability data are based on statistics of large number devices. More clear description on stability tests has been added in the revised manuscript. See page 18, line 4 to page 19, line1, highlighted by the yellow color.

Q2: The 85 °C thermal stability test conditions should be specified here, such as open circuit, short circuit or Vmpp conditions.

Responses: Thanks for the comments. For 85 °C thermal aging, the devices were aging in the dark without load (open circuit). Related description has been added in the revised manuscript. Please see page 18, line 12-13, highlighted by the yellow color.

Q3: The conductivity of Bi is not comparable to FTO, otherwise, your Bi thickness would not be limited at 20 nm. The enhancement of R_s and drop of FF is obvious when the thickness goes to 40 nm. In fact, the conductivity of Bi layer one order of magnitude lower than regular FTO.

Responses: Thanks for reminding us to deeply think about this point. Referee 1 also has raised a similar question (see **Q5** of referee 1). I copy the same answer here.

This time, we have carefully checked the Bi film's crystal growth mechanism, and we have found the inherent reason leading to the abnormally increased series resistance.

As shown in the SEM and AFM images below (as well as in the revised **Supplementary Figure S8**), when the thickness of the Bi films increases from 10 nm, 40 nm to 80 nm, the surface roughness changes significantly from 3.7 nm, 12.3 nm to 37.7 nm. Some tower-shaped protuberances with the height of ~300 nm could be detected in the 80 nm-thick Bi film. We think the high roughness of Bi film will lead to an increased contact resistance between Bi and Ag, and therefore a notably increased series resistance of the solar cell (similar phenomenon has been reported in the reference: Schubert, S., Meiss, J., Müller-Meskamp, L., & Leo, K.. Improvement of transparent metal top electrodes for organic solar cells by introducing a high surface energy seed layer. *Adv. Energy Mater.* **3**, 201200903 (2013)). Related description has been added in the revised manuscript. Please see page 13, line 20 to page 14, line 4, highlighted by the yellow color.

Supplementary Figure S8. AFM (scale bar: 1 μm) and SEM (scale bar: 500 nm) cross-sectional images of the Bi layers (onto the $\text{MAPbI}_3/\text{PCBM}/\text{BCP}$) with the thickness of 10 nm (**a, d**), 40 nm (**b, e**), and 80 nm (**c, f**).

By the way, the sheet resistant of 80 nm-thick Bi film is measured to be 55 Ohms/sq., while that of 500 nm-thick FTO glass is 8.4 Ohms/sq. (see the images shown below). Therefore, it is reasonable for us to mention that the Bi film has a competitive conductivity to that of ITO/FTO.

Sheet resistances of 80 nm-thick Bi film and FTO glass measured by Four-point probe resistance meter

Reviewer #3:

Review of Wu et al. "A chemically inert bismuth interlayer enhances long-term stability of inverted perovskite solar cells"

The work by Wu et al. presents a quite compelling device study on the use of a bismuth metallic interlayer in an inverted perovskite photovoltaic device. Although much of the data is of considerable interest the insight provide by the work is somewhat limited and precludes the reviewer from recommending it for publication in Nature communications. The use of the Bi metal is somewhat novel bulk of the work carried out on MAPbI₃ with targeted demonstrations of mix cations appears well conceived and intersting so if the following issues are addressed comprehensively it might become suitable for publication.

Q1: Specifically the authors fail to cite the relevant ISOS protocols used for the device stability testing or reference the relevant protocols (doi:10.1016/j.solmat.2011.01.036). More serious is the providing of a rational for the test protocols that are employed. The rational for shelf life test vs the 85C N₂ testing in the dark in comparison to 45C N₂ testing under continuous illumination is not provided by the authors. While it is clear that the inclusion of Bi improves the devices the article provides little insight as to why this strategy is succesfull or indeed why it maybe succesfull and thus the impact of the work is significantly reduced as how this might be translated by other researchers is unclear. This is exemplified by the lack of a clear rational for the testing mentioned above and although there maybe a reasonable rational for these test it is not provided within the manuscript. This issues must be addressed given the results by Domanski et al. (<https://doi.org/10.1038/s41560-017-0060-5>) which should be referenced. Data presented in the SI are in many ways more compelling as it suggest a more general set of design rules and a overarching framework for the choice of metal interlayers which might be less susceptible to chemical instabilities within the device stack.

Responses: Thanks very much for the important comments and constructive suggestions. In the revised manuscript, we have put attentions on rationality of the experimental designs and results.

- (1) The rational for the aging test protocols has been given in the first paragraph of “**Stability tests of the devices**”. The suggested literature 1 ([doi:10.1016/j.solmat.2011.01.036](https://doi.org/10.1016/j.solmat.2011.01.036)) has been cited in the revised manuscript as reference 48 at page 18, line 7, highlighted by the yellow color. (Reference 48. Reese, M. O. et al. Consensus stability testing protocols for organic photovoltaic materials and devices. *Solar Energy Mater. Solar Cells* **95**,

1253–1267 (2011)). The literature 2 ([doi:10.1038/s41560-017-0060-5](https://doi.org/10.1038/s41560-017-0060-5)) has been cited in the revised manuscript as reference 52 at page 18, line 7, highlighted by the yellow color. (Reference 52. Domanski, K., Alharbi, E. A., Hagfeldt, A., Grätzel, M. & Tress, W. Systematic investigation of the impact of operation conditions on the degradation behaviour of perovskite solar cells. *Nat. Energy* **3**, 61–67 (2018)). See below is the copied part in the revised manuscript (Also see page 18, line 4 to page 19, line 1, highlighted by the yellow color.):

“The excellent anti-corrosive property and robust impermeability of the introduced Bi interlayer clearly demonstrated in the above discussion may largely improve the stability of HPVSKSCs. The stability of devices should be assessed under specified aging conditions, which have been well discussed in recent review papers.^{48,50,52} A storage stability test was conducted in the dark in ambient air at room temperature (RT) without humidity control (relative humidity of the local climate is normally 40-90%). Separately, a thermal aging test was implemented at 85 °C as specified for the aging temperature in the standard IEC 61215-1: 2016 (Terrestrial Photovoltaic (PV) Modules - Design Qualification and Type Approval - Part 1: Test Requirements).⁴⁹ As proposed by Snaith *et al.*,⁵⁰ our thermal aging was conducted in the dark in an inert atmosphere (N₂) filled glovebox for eliminating the interference from other stresses, such as H₂O, O₂, and light. All devices were open circuits during thermal aging. Also, the operational stability was measured at an elevated temperature (45 °C) with the devices biased near their MPPs by referring to the standard IEC 61646: 2008 (Thin-film Terrestrial Photovoltaic (PV) Modules—Design, Qualification and Type Approval).⁵¹ As Grätzel *et al.* advocated,⁵² to eliminate interference from ultraviolet light and stress from H₂O, and O₂, our light soaking test was performed in a N₂ filled glovebox by using a white light LED array as the light source. The light intensity was calibrated to achieve the same J_{SC} from the HPVSKSCs as for 1 sun AM1.5G solar irradiation.”

- (2) The reasons why inclusion of Bi improves the devices' stability have been clearly discussed by adding new XPS and XRD results. See the part of “**Chemically inert characteristic of Bi film**” from page 7, line 9 to page 10, line 8 in the revised manuscript and the added XPS, XRD data shown in the revised **Figure 1**, **Supplementary Figure S1**, **Supplementary Figure S2** and **Supplementary Figure S3**. Through the new experiments, it is very clear that in comparison to Ag, Cu and Au, only Bi is unique, because it is anticorrosive to halides perovskites and will not lead to electronic contamination to perovskites by interfacial diffusion. A thermodynamic analysis providing insight into the reaction mechanism has been in the revised **Figure 1b**. A schematic diagram of metal diffusion into the perovskite

has been highlighted in the revised **Figure 1c**. These may benefit the readership to quickly catch the main point of our work. The added **Figure 1** is copied and shown below for easier read:

Figure 1 | Chemical reaction of metal-perovskite. (a) XPS spectra of Pb 4f for fresh and aged samples, including MAPbI₃/Bi, MAPbI₃/Au, MAPbI₃/Cu, MAPbI₃/Ag and only MAPbI₃ (aging conditions: in the dark, in N₂, 85 °C, 100 h). (b) Schematic diagram and the corrosion reaction equations. The standard enthalpy of formation ($\Delta_f H$, 298.15 K) and the reaction barrier (ΔE) are also listed. (c) Schematic diagram of metal diffusion into the perovskite and the atomic ratios at the upper surface.

Q2: It is notable that other important work in stability is also not referenced particularly work by NREL (DOI:10.1021/acsenergylett.6b00013,https://doi.org/10.1038/s41560-017-0067-y) or on the chemistry of the fulleren based interlayers used in many composite by researchers at the Univ of Washington (notably DOI: 10.1002/adma.201300580) which indicates the types of chemistries likely playing an important role at these functional device interfaces as well as this groups more recent work directly in perovskite solar cells. Similarly the question of the metallic interfaces analogs and associated oxide literature is also not well referenced (DOI: 10.1021/acсами.6b10898 and DOI: 10.1021/acsenergylett.6b00320) with the later appearing to be directly relevant to the work reported here and providing a framework that could rationalize the material choice and its stability performance. The issue of metal ingress generally has also been addressed by McGehee group with recent work on stable cell architectures and packages, the referee would suggest the implicates of the presented work in the context of these recent publications would increase the impact of this work.

Responses: Thank you very much for your comments. We have cited the above literatures reasonably. Details are as follows.

1. DOI:10.1021/acsenergylett.6b00013: This literature has been cited in the revised manuscript as reference 18 at page 4, line 10, highlighted by the yellow color.
Reference 18. Sanehira, E. M. *et al.* Influence of electrode interfaces on the stability of perovskite solar cells: reduced degradation using MoO_x/Al for hole collection. *ACS Energy Lett.* **1**, 38-45 (2016).
2. https://doi.org/10.1038/s41560-017-0067-y: This literature has been cited in the revised manuscript as reference 20 at page 4, line 10, highlighted by the yellow color.
Reference 20. Christians, J. A. *et al.* Tailored interfaces of unencapsulated perovskite solar cells for > 1,000 hour operational stability. *Nat. Energy* **3**, 68-74 (2018).
3. on the chemistry of the fulleren based interlayers used in many composite by researchers at the Univ of Washington: This literature has been cited in the revised manuscript as reference 28 at page 4, line 16, highlighted by the yellow color.
Reference 28. Yao, K. *et al.* Fullerene-anchored core-shell ZnO nanoparticles for efficient and stable dual-sensitized perovskite solar cells. *Joule* (2018) (DOI: 10.1016/j.joule.2018.10.018)
4. DOI: 10.1021/acсами.6b10898: This literature has been cited in the revised manuscript as reference 19 at page 4, line 9, highlighted by the yellow color.
Reference 19. Schulz, P. *et al.* High-work-function molybdenum oxide hole extraction contacts in hybrid organic-inorganic perovskite solar cells. *ACS Appl. Mater. Interfaces* **8**, 31491-31499 (2016).

5. DOI: 10.1021/acsenergylett.6b00320: This literature has been cited in the revised manuscript as reference 32 at page 5, line 20, highlighted by the yellow color.

Reference 32. Zhao, L. et al. Redox chemistry dominates the degradation and decomposition of metal halide perovskite optoelectronic devices. *ACS Energy Lett.* **1**, 595–602 (2016).

6. The issue of metal ingress generally has also been addressed by McGehee group with recent work on stable cell architectures and packages: This literature has been cited in the revised manuscript as reference 14 at page 4, line 9, highlighted by the yellow color.

Reference 14. Boyd, C. C. et al. Barrier design to prevent metal-induced degradation and improve thermal stability in perovskite solar cells. *ACS Energy Lett.* **3**, 1772-1778 (2018).

Q3: The above issues are the most critical to be addressed as if done so can address the confusion that currently exists within the paper regarding what the Bi layer is actually doing. The current confusion is result of the authors not providing a clear picture or posing a clear hypothesis of what is actually happening. At present multiple hypotheses appear but without any really being tested comprehensively leaving the reader with simply the device data and that the Bi based systems appear to be a bit better.

It appears to this reviewer that in the context of the above literature the existing data can be better understood and this understanding presented given with the exiting data. Specifically the criteria of anti-corrosive metals and the associated deposition conditions should be expanded on. Related to this is the use of the description of "anti-corrosive" evaporation and metal systems in the context of this device work. It is the referees perspective that the associated discussion is perhaps one of the most compelling aspects of the work, corrosion appears to be an accurate general descriptor but should the existing draft does not provide enough specificity regarding the actual corrosion process/chemistry in this case. While the reviewer understands the choice of examining the PCBM/BCP/Bi/Ag and related layers it would be instructive if to examine the MAPbI₃/Bi layers and associated material morphology, structure and interfacial chemistry to understand the potential advantages of the Bi interface.

Responses: Thank you very much for the important comments and constructive suggestions. It is thought **Q3** partially overlaps with **Q1**. We explain a little bit more as follows.

In the revised manuscript, we have carefully revised **Figure 1** in order to give a clear picture on what is actually happening and why it is important by introducing Bi in HPVKSC. The important XPS new data, in combination with a thermodynamic analysis and a schematic diagram depicting the mechanism have been given in the revised **Figure 1**. The compared metals have been expanded to the popularly used

electrode materials including Ag, Cu, Au. Regarding to the adverse corrosion reaction, MAPbI₃/metal samples were prepared by evaporating 5-nm Bi, Cu, Au or Ag films directly onto MAPbI₃, and then XPS was implemented for the fresh and aged samples (aging conditions: dark, N₂, 85 °C for 100 h). Solid evidences in XPS and XRD data (see **Figure 1**, **Supplementary Figure S1**, **Supplementary Figure S2** and **Supplementary Figure S3**) consistently indicate that Bi is unique in comparison to other metals. Not like other metals, there is no corrosion reaction and limited metallic diffusion occurring between halide perovskite and Bi even under harsh conditions. Detail description has been given in the part of “**Chemically inert characteristic of Bi film**” from page 7, line 9 to page 10, line 8, highlighted by the yellow color in the revised manuscript. Hopefully, these revisions could clarify the working mechanism of Bi interlayer, and convince the reviewers and the possible readership that the first time introduced Bi in this work is important in HPVKSCs.

Q4: Other minor issues which should also be addressed. As an example perovskite solar cells (PVKSC) is probably not a sufficiently specific designation (nor is the other common designation of PSC currently often used for perovskite solar cells as it is also commonly used for plastic solar cells) as this can also refer to the oxide systems which have been demonstrated but operate on bulk photovoltaic effect and leverage ferroelectric behaviors. The existing work would seem to apply to any of the halide perovskite solar cell systems and some designation reflecting this should be used (i.e. HPVSKSC or HPSC for halide perovskite solar cells systems which the work targets).

Responses: Thanks for the comments. We have changed the description in the revised manuscript according to the reviewer’s suggestion. We have changed “perovskite solar cells” and “PVKSC” to the sufficiently specific designation “halide perovskite solar cell” and “HPVKSC”, respectively.

Q5: Similarly the data in figure 3 (TOF-SIMS) appears to have the halide signal (I-) completely saturated and no clear indication of the polarity (while this can be inferred from the data). Although these issues with the TOF-SIMS doesn't undercut the insight from these measurements these details should be provided in a more complete way (see recent review on this technique for halide perovskite solar cells DOI:10.1021/acsami.8b07937).

Responses: Thank you for your suggestion. In the revised **Figure 4**, I₂⁻ has been given as a supporting data for I⁻, since the intensity of I⁻ ions is too strong and beyond the upper detecting limit. And the I₂⁻ signal has the same tendency to that of I⁻.

Figure 4 | Elemental distribution in HPVKSCs. ToF-SIMS elemental depth profiles of MA-HPVKSCs. (a) and (c) are the fresh devices with and without a Bi interlayer with a thickness of 20 nm; (b) and (d) are the devices aged at 85 °C for 100 h in the dark under N₂ atmosphere.

REVIEWERS' COMMENTS:

Reviewer #1 (Remarks to the Author):

The authors have revised their paper. It may now be accepted.

Reviewer #3 (Remarks to the Author):

The authors have done an excellent job in improving the manuscript on many fronts. As a result of this significant effort it is this reviewers opinion that the current version is clear and well executed. While questions regarding the Li content of the HTM persist in this reviewers mind it does not undercut the science and technical results presented. As a result of the work it is the opinion of the reviewer that the work of the standard and impact that warrant publication in nature communications.

REVIEWERS' COMMENTS:

Reviewer #1 (Remarks to the Author):

The authors have revised their paper. It may now be accepted.

Reviewer #3 (Remarks to the Author):

The authors have done an excellent job in improving the manuscript on many fronts. As a result of this significant effort it is this reviewer's opinion that the current version is clear and well executed. While questions regarding the Li content of the HTM persist in this reviewer's mind it does not undercut the science and technical results presented. As a result of the work it is the opinion of the reviewer that the work of the standard and impact that warrant publication in nature communications.